# ClavaDDPM: Multi-relational Data Synthesis with Cluster-guided Diffusion Models

Wei Pang[1,2], Masoumeh Shafieinejad[2], Lucy Liu[3], Stephanie Hazlewood[3], and Xi He [1,2*]

[1]University of Waterloo
[2]Vector Institute
[3]Royal Bank of Canada
w3pang@uwaterloo.ca, masoumeh@vectorinstitute.ai, lucy.z.liu@rbc.com,
stephanie.hazlewood@rbc.com, xi.he@uwaterloo.ca

## Abstract

Recent research in tabular data synthesis has focused on single tables, whereas real-world applications often involve complex data with tens or hundreds of interconnected tables. Previous approaches to synthesizing multi-relational (multi-table)[2] data fall short in two key aspects: scalability for larger datasets and capturing long-range dependencies, such as correlations between attributes spread across different tables. Inspired by the success of diffusion models in tabular data modeling, we introduce **C**luster **La**tent **Va**riable guided **D**enoising **D**iffusion **P**robabilistic **M**odels (ClavaDDPM). This novel approach leverages clustering labels as intermediaries to model relationships between tables, specifically focusing on foreign key constraints. ClavaDDPM leverages the robust generation capabilities of diffusion models while incorporating efficient algorithms to propagate the learned latent variables across tables. This enables ClavaDDPM to capture long-range dependencies effectively. Extensive evaluations on multi-table datasets of varying sizes show that ClavaDDPM significantly outperforms existing methods for these long-range dependencies while remaining competitive on utility metrics for single-table data.

## 1 Introduction

**Motivation.** Synthetic data has attracted significant interest for its ability to tackle key challenges in accessing high-quality training datasets. These challenges include: i) data scarcity [14, 53], ii) privacy [2, 19], and iii) bias and fairness [46]. The interest in synthetic data has extended to various commercial settings as well, notably in healthcare [18] and finance [36] sectors. The synthesis of tabular data, among all data modalities, is a critical task with approximately 79% of data scientists working with it on a daily basis [45]. While the literature on tabular data synthesis has predominantly focused on single table (relation) data, datasets in real-world scenarios often comprise multiple interconnected tables and raise new challenges to traditional single-table learning [38, 3, 12, 22]. These challenges have even enforced a join-as-one approach [15, 17], where the multi relations are first joined as a single table. However, with more than a couple of relations (let alone tens or hundreds of them as in the finance sector) this approach is neither desirable nor feasible.

**Challenges.** Synthetic Data Vault [35] and PrivLava [5] are recent efforts to synthesize multi-relational data using hierarchical and marginal-based approaches. These methods exhibit significant

---

[*]Corresponding author.
[2]In the context of databases, a ***table*** is formally referred to as a ***relation***. Throughout this work, we use these terms interchangeably.

limitations in processing speed and scalability, both with respect to the number of tables and the domain size of table attributes, and they often lack robustness in capturing intricate dependencies. Alternatively, diffusion models have emerged as powerful tools for data synthesis, demonstrating remarkable success in various domains [37]. These models are particularly noted for their strong capabilities in controlled generation. Despite their potential, the application of diffusion models to tabular data synthesis has been limited to unconditional models [25, 50, 28, 24], leaving a gap in effectively addressing the multi-table synthesis problem.

**Solution.**    To address these challenges, we introduce ClavaDDPM (Cluster Latent Variable guided Denoising Diffusion Probabilistic Models). Our novel approach leverages the controlled generation capabilities of diffusion models by utilizing clustering labels as intermediaries to model the relationships between tables, focusing on the foreign-key constraints between parent and child tables. This integration of classifier guidance within the diffusion framework allows ClavaDDPM to effectively capture complex multi-table dependencies, offering a significant advancement over existing methods.

**Contributions.**    In this work, we: 1) provide a complete formulation of the multi-relational modeling process, as well as the essential underlying assumptions being made, 2) propose an efficient framework to generate multi-relational data that preserves long-range dependencies between tables, 3) propose relationship-aware clustering as a proxy for modeling parent-child constraints, and apply the controlled generation capabilities of diffusion models to tabular data synthesis, 4) apply an approximate nearest neighbor search-based matching technique, as a universal solution to the multi-parent relational synthesis problem for a child table with multiple parents, 5) establish a comprehensive multi-relational benchmark, and propose *long-range dependency* as a new metric to measure synthetic data quality specific to multi-table cases, and 6) show that ClavaDDPM significantly outperforms existing methods for these long-range dependency metrics while remaining competitive on utility metrics for single-table data.

## 2   Related work

**Single-table synthesis models.**    Bayesian network [48] is a traditional approach for synthetic data generation for tabular data. They represent the joint probability distribution for a set of variables with graphical models. CTGAN [47] is a tabular generator that considers each categorical value as a condition. CTAB-GAN [52] includes mixed data types of continuous and categorical variables. Several studies have explored how GAN-based models can contribute to fairness and bias removal [44, 45]. In privacy, GAN-based solutions boosted with differential privacy have not been as successful as their Baysian-network-based competitors [34, 51]. Recent popular Diffusion Models, [20, 40, 42, 41], offer a different paradigm for generative modeling. TabDDPM [25] utilizes denoising diffusion models, treating numerical and categorical data with two disjoint diffusion processes. STaSy [24] uses score-based generative modeling in its training strategy. CoDi [28] processes continuous and discrete variables separately by two co-evolved diffusion models. Unlike the previous three which perform in data space, TabSyn [50] deploys a transformer-based variational autoencoder and applies latent diffusion models. Privacy and fairness research for diffusion models are currently limited to a few studies in computer vision [26, 11, 16].

**Multi-table synthesis models.**    There have been few proposals for synthetic data generation for multi-relational data. A study proposed this synthesis through graph variational autoencoders [31], the presented evaluation is nevertheless very limited. The Synthetic Data Vault [35] uses the Gaussian copula process to model the parent-child relationship. SDV iterates through each row in the table and performs a conditional primary key lookup in the entire database using the ID of that row, making a set of distributions and covariance matrices for each match. This inhibits an efficient application of SDV to the numerous tables case. PrivLava [5], synthesizes relational data with foreign keys under differential privacy. The key idea of PrivLava is to model the data distribution using graphical models, with latent variables included to capture the inter-relational correlations caused by foreign keys.

## 3   Background

**Multi-relational databases.**    A multi-relational database $\mathcal{R}$ consists of $m$ tables (or relations) $(R_1, \ldots, R_m)$. Each table is a collection of rows, which are defined over a sequence of attributes.

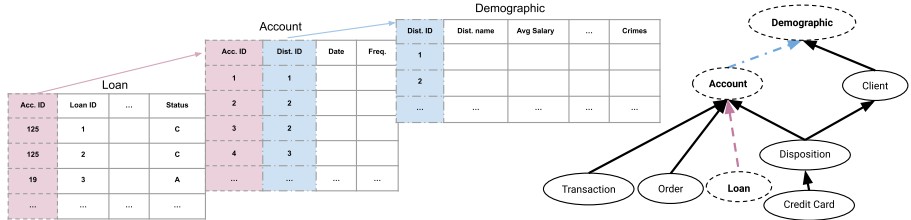

Figure 1: *Berka* sample tables (left), and the foreign key constraint graph for *Berka* (right)

One of the attributes, let's consider the first attribute without loss of generality, is the **primary key** of table $R$, which serves as the unique identifier for each row in the table. No rows in the same table have repeated values for the primary key attribute. We use *Berka* database [4] as our running example in this work, as in Figure 1. Note the *Account ID*, the primary key for the *Account* table in *Berka*.

Given a table $R_j$, we say a relation $R_i$ has a **foreign key constraint** with $R_j$, or $R_i$ **refers to** $R_j$, if $R_i$ has an attribute known as **foreign key** that refers to the primary key of $R_j$: for every row $r_i \in R_i$, there exists a row $r_j \in R_j$ such that $r_j$'s primary key value equals to $r_i$'s foreign key value. For example, the *Account ID* of the *Loan* table refers to the primary key of the *Account* table. If an account row is removed from the *Account ID* table, so would all the referring rows in the *Loan* table to this account, for foreign key constraint to hold. Note that the primary key of a table can consist of multiple attributes. In this paper, we focus on the case of a single attribute that is common in practice. Also note that all keys are considered row identifiers and are thus not treated or modeled alongside the actual table attributes in this work.

A multi-relational database under foreign key constraints forms a directed acyclic graph (DAG),

$$\mathcal{G} = (\mathcal{R}, \mathcal{E}), \mathcal{E} = \{(R_i \to R_j) \mid i, j \in \{1, \dots, m\}, i \neq j, R_i \text{ refers to } R_j\} \quad (1)$$

with the tables $\mathcal{R}$ being the set of nodes, and $\mathcal{E}$ being the set of edges. In addition, for $R_i$ referring to $R_j$, we also call this a **parent-child relationship**, where $R_j$ is the **parent** and $R_i$ is the **child**. We use the **maximum depth** to denote the number of nodes on the longest path in $\mathcal{G}$. Figure 1 shows the corresponding graph to *Berka* database and its maximum depth is 4.

**Multi-relational synthesis problem.** Given a multi-relational database $\mathcal{R} = \{R_1, \dots, R_m\}$, we would like to generate a synthetic version $\tilde{\mathcal{R}} = \{\tilde{R}_1, \dots, \tilde{R}_m\}$ that has the same structure and foreign-key constraints as $\mathcal{R}$ and preserves attribute correlations within $\mathcal{R}$, including 1) the inter-column correlations within the same table; 2) the intra-group correlations within the same foreign key group; 3) the inter-table correlations. The first aspect has been well defined, measured, and tackled in the literature of single-table synthesis [52, 25, 50] while the other two aspects are raised due to foreign-key constraints between tables [5]. For instance, in *Berka* database (Figure 1), the foreign key constraint between the *Loan* table and the *Account* table via *Account ID* adds an important intra-group correlation for the combinations of loans associated with an account and many 1-hop inter-table correlations between columns in the *Loan* table and the columns in the *Account* table. Even for the *Loan* table and the *Demographic* table that are indirectly constrained by foreign keys, their columns are correlated as well, e.g., how is the average salary in a district related to the status of loans, an example for 2-hop inter-table correlation.

**Classifier-guided DDPM.** DDPM [20] uses two Markov chains, a forward chain that perturbs data to noise through a series of Gaussian transitions, and a reverse chain that converts noise back to data with the same number of steps of Gaussian transitions (Equation 2).

$$q\left(\boldsymbol{x}_t \mid \boldsymbol{x}_{t-1}\right) \coloneqq \mathcal{N}\left(\boldsymbol{x}_t; \sqrt{1 - \beta_t}\boldsymbol{x}_{t-1}, \beta_t \boldsymbol{I}\right)$$
$$p_\theta\left(\boldsymbol{x}_{t-1} \mid \boldsymbol{x}_t\right) \coloneqq \mathcal{N}\left(\boldsymbol{x}_{t-1}; \boldsymbol{\mu}_\theta\left(\boldsymbol{x}_t, t\right), \boldsymbol{\Sigma}_\theta\left(\boldsymbol{x}_t, t\right)\right). \quad (2)$$

Prior work [40] shows that given label $\boldsymbol{y}$, the conditional reverse process has the form

$$p_{\theta,\phi}\left(\boldsymbol{x}_t \mid \boldsymbol{x}_{t+1}, \boldsymbol{y}\right) \propto p_\theta\left(\boldsymbol{x}_t \mid \boldsymbol{x}_{t+1}\right) p_\phi\left(\boldsymbol{y} \mid \boldsymbol{x}_t\right). \quad (3)$$

By approximating $\log p_\phi(\boldsymbol{y} \mid \boldsymbol{x}_t)$ using Taylor expansion around $\boldsymbol{x}_t = \boldsymbol{\mu}$, the conditional reverse process (Equation 3) can be approximated with a perturbed Gaussian transition [10]

$$\log\left(p_{\theta,\phi}\left(\boldsymbol{x}_t \mid \boldsymbol{x}_{t+1}, \boldsymbol{y}\right)\right) \approx \log\left(p\left(\boldsymbol{z}\right)\right) + C, \ \ \boldsymbol{z} \sim \mathcal{N}\left(\boldsymbol{\mu} + \boldsymbol{\Sigma g}, \boldsymbol{\Sigma}\right), \tag{4}$$

where $C$ is a constant and $\boldsymbol{g} = \nabla_{\boldsymbol{x}_t} \log\left(P_\phi\left(\boldsymbol{y} \mid \boldsymbol{x}_t\right)\right)\big|_{\boldsymbol{x}_t = \boldsymbol{\mu}}$ computed from the classifier $P_\phi$.

# 4 ClavaDDPM

Here, we elaborate on the training and synthesis process of ClavaDDPM, and each design's rationale.

## 4.1 Modeling generative process for two-table relational databases

**Notations.** Consider a database of two tables $\mathcal{R} = \{R_1, R_2\}$, e.g. {*Loan, Account*} in *Berka*, where the child table $R_1$ refers to parent table $R_2$. To model the entire database, we first use $\boldsymbol{X}$ and $\boldsymbol{Y}$ as the variables for the child table $R_1$ and parent table $R_2$, respectively (dropping their primary key attributes and indexing their respective row variables starting from one). In this section, we use boldface to represent random variables. e.g. $Y \sim \boldsymbol{Y}$, where $Y$ is the data of $R_2$, and $\boldsymbol{Y}$ is the random variable $Y$ being sampled from. In addition, we use subscript to represent the ***parent row*** some data or random variable refers to. e.g. $\boldsymbol{x}_j$ represents the child random variable who refers to parent $\boldsymbol{y}_j$. Refer to Appendix A for a complete list of notations used and the corresponding design choices.

**Assumptions.** 1) The parent table has no constraints itself. Hence, we can follow previous work on single-table synthesis [13, 25, 47, 50, 52] to make an i.i.d assumption on the rows in the parent table. The parent table $\boldsymbol{Y}$ can be modeled as a list of i.i.d. row variables $\{\boldsymbol{y}_j \mid j = 1, \ldots, |R_2|\}$, where $j$ is the index or the primary key value of the $j$th row, and each row follows a distribution $p(y)$.

2) The i.i.d assumption does not apply to the child table rows ($\boldsymbol{x}_j$'s) as they are constrained by their respective parent rows. Consider two loans associated with the same account id; if one's status is *in debt* ("C"), the other one is likely so too. To capture this dependency, we make a Bayesian modeling assumption that, although child rows associated with the same parent row are not independent, they are conditionally independent of child rows associated with other parent rows, given their respective parent. For example, consider an *account* table (parent) and a *loan* table (child). Loans related to the same account (i.e., the same parent) are not independent due to shared account-specific factors. However, loans from different accounts can be considered conditionally independent when accounting for their respective account-level information. Hence, we model $\boldsymbol{X}$ by $\{\boldsymbol{g}_j \mid j = 1, \ldots, |R_2|\}$, where each group $\boldsymbol{g}_j = \{\boldsymbol{x}_j^i \mid i = 1, \ldots, |\boldsymbol{g}_j|\}$ represents a set of child table rows referring to the parent row $\boldsymbol{y}_j$.

3) Without violating the assumptions made above, we further make an i.i.d assumption on $(\boldsymbol{g}_j, \boldsymbol{y}_j)$, which leads to an approximated distribution for the parent-child tables:

$$P(\boldsymbol{X} = X, \boldsymbol{Y} = Y) \approx \prod_{j=1}^{|R_2|} P\left(\boldsymbol{g}_j = g_j, \boldsymbol{y}_j = y_j\right) \quad \text{or} \quad p(X, Y) = \prod_j p(g_j, y_j) \tag{5}$$

where $X = \cup_{j=1}^{|R_2|} g_j$ and $g_j = \{x_j^1, \ldots, x_j^{|g_j|}\}$. This model allows us to capture the inter-table correlations (the correlation between tuples from different tables) and the intra-group correlations.

**Modeling.** Despite the simplified formulation with several aforementioned assumptions, learning the distribution $p\left(g_j, y_j\right)$ is non-trivial. In particular, $(g_j, y_j)$ cannot be flattened into a matrix form for learning since the set structured attributes in $\boldsymbol{g}_j$, e.g., the size of a group variable $\boldsymbol{g}_j$ is not fixed.

A naive solution is to model a conditional distribution of the group given the parent row

$$p\left(g_j, y_j\right) = p\left(g_j \mid y_j\right) p\left(y_j\right) \tag{6}$$

Direct modeling of Equation (6) still has the same issue as before for the foreign key group $\boldsymbol{g}_j$, which can take an arbitrary number of child rows. In particular, when modeling $\boldsymbol{g}_j = f\left(\boldsymbol{x}_j\right)$ for some function $f$, there is no trivial structured support for $\boldsymbol{g}_j$ if we model for $\boldsymbol{g}_j$ using only the attributes or features of the child rows. Furthermore, the conditioning space of the parent row $\boldsymbol{y}$ can be very

large (e.g., *Account* table has a domain size of more than 11,000), which can lead to poorly learned conditional distribution if we treat $\boldsymbol{y}$ as labels in the classifier-guided DDPM. The original space of $\boldsymbol{y}$ is high-dimensional and noisy and does not guarantee any spatial proximity or smoothness. In the context of deep modeling, this drastically worsens the quality of conditional sampling.

To address these two challenges, we introduce latent random variables $\boldsymbol{c}$ such that $\boldsymbol{g}_j$ is independent from $\boldsymbol{y}_j$ conditioned on $\boldsymbol{c}$

$$\boldsymbol{g}_j \perp\!\!\!\perp \boldsymbol{y}_j \mid \boldsymbol{c} \tag{7}$$

With this assumption, we get an indirect modeling of inter-table correlations through $\boldsymbol{c}$:

$$p(g_j, y_j) = \sum_c p(g_j, y_j|c)p(c) = \sum_c p(g_j|c)p(y_j|c)p(c) = \sum_c p\left(g_j \mid c\right) p\left(y, c\right) \tag{8}$$

Compared to Equation (6), when $\boldsymbol{c}$ is selected to be lying on a low-dimensional, compact manifold, the latent conditional distribution $p\left(g_j \mid c\right)$ will be easier to model than $p\left(g_j \mid y_j\right)$.

For each foreign key group $\boldsymbol{g}_j$, we model its size explicitly with a variable $\boldsymbol{s}_j$. By making an assumption that $\boldsymbol{s}_j$ is conditionally independent from its child row variables $\{\boldsymbol{x}_j^1, \ldots, \boldsymbol{x}_j^{s_j}\}$ given the latent random variable $\boldsymbol{c}$, we essentially defined a generative process for $g_j$: first sample its size $s_j \sim \boldsymbol{s}_j$, then sample $g_j$ child row variables. In addition, we make an i.i.d assumption on the child row variables given the latent variable. Hence, we have

$$p(g_j|c) = p(s_j|c) \prod_{i=1}^{s_j} p(x_j^i|c) \tag{9}$$

Putting all together, we have the final formulation of the generative process for a two-table case:

$$
\begin{aligned}
p\left(X, Y\right) &\approx \prod_{j=1}^{|R_2|} p\left(g_j, y_j\right) && \text{i.i.d assumption on } \left(\boldsymbol{g}_j, \boldsymbol{y}_j\right) \text{ Equation (5)} \\
&\approx \prod_{j=1}^{|R_2|} \sum_c p(s_j|c) \prod_{i=1}^{s_j} p(x_j^i|c)p(y_j, c) = \prod_{j=1}^{|R_2|} \sum_c p\left(y_j, c\right) p\left(s_j \mid c\right) \prod_{i=1}^{s_j} p\left(x_j^i \mid c\right).
\end{aligned}
\tag{10}
$$

Unlike our naive method that uses a direct modeling of $p\left(x_j \mid y_j\right)$, we model $p\left(x_j \mid c_j\right)$, which greatly reduced the condition space, thus better capturing the inter-table correlation between $\boldsymbol{X}$ and $\boldsymbol{Y}$. On the other hand, the intra-group correlations are intrinsically addressed, because our modeling of $\left(\boldsymbol{g}_j, \boldsymbol{y}_j\right)$ and the corresponding dependency assumptions enforce that two child rows are drawn from the same distribution if and only if they belong to the same foreign key group.

Based on Equation (10), we introduce our generative process for two-table case.

**Phase I: Latent learning and table augmentation**: (1) Learn latent variable $c$ on the joint space $(X; Y)$, such that each parent row $y_j$ corresponds to a learned latent variable $c_j$. (2) Augment the parent table into $T_Y = (Y; C)$, where $C$ corresponds to the latent variable values $c_j$ for each row $y_j$.

**Phase II: Training**: (3) Train diffusion model $p_\theta\left(y, c\right)$ on augmented table $T_Y$ and child diffusion model $p_\phi\left(x\right)$. (4) Given learned latents and child table, train classifier $p_\psi\left(c \mid x\right)$. (5) Estimate the foreign key group size distributions conditioned on latent variables $p\left(s \mid c\right)$.

**Phase III: Synthesis**: (6) Synthesize the augmented parent table $\tilde{T}_Y = (\tilde{Y}; \tilde{C}) \sim p_\theta\left(\cdot, \cdot\right)$. (7) For each synthesized latent variable $\tilde{c}_j \in C$, sample group size $\tilde{s}_j \sim p\left(\cdot \mid \tilde{c}_j\right)$. (8) Given $\tilde{s}_j$, sample each child row within the foreign key group $x_j^i \sim p_{\phi,\psi}\left(\cdot \mid \tilde{c}_j\right)$, where $p_{\phi,\psi}$ performs classifier guided sampling by perturbing $p_\phi$ with the gradient of $p_\psi$. We denote steps (7) and (8) by $\tilde{X} \sim p(\cdot|C)$.

## 4.2 Extension to more parent-child constraints

We learn the latent variables between a parent and a child pair in a bottom-up fashion (starting from the leaf nodes in $\mathcal{G}$) and pass all the latent variable values to the parent table for the next set of latent variables at higher levels. Given a parent-child pair $(Y, X)$, the child table $X$ also

has $k$ leaf node children, $Z_1, \ldots, Z_k$. Let $c_{X,Z_i}$ represent the latent variables learned on the joint space $(X; Z_i)$. The augmented table for $X$ is formed by appending all its latent variable values, i.e., $T_X = (X; C_{X,Z_1}; \ldots; C_{X,Z_k})$. Then, the latent variable $c_{Y,X}$ is learned on the joint space of $(Y; T_X)$ instead of $(Y; X)$. Therefore, our latent learning process follows a bottom-up topological order, ensuring each child table is already augmented by the time we learn the latent variable to augment its parent.

The training phase and the synthesis phase are similar to the two-table case, by handling the parent-child tables in a top-down topological order using the augmented tables. We detail the end-to-end algorithms for the complex data in Appendix B. However, we would like to highlight a special case when a table $X$ has ***multiple parents*** $Y_1, \ldots, Y_k$. During synthesis, we will have $k$ synthetic latent variable $\tilde{C}_1, \ldots, \tilde{C}_k$ corresponding to the $k$ parents, and thus $k$ copies of synthetic child tables $\tilde{X}_1 \sim p(\cdot \mid \tilde{C}_1), \ldots, \tilde{X}_k \sim p(\cdot \mid \tilde{C}_k)$. Unifying these diverged synthetic tables presents a challenge and we present a universal solution in Section 4.3.3.

Extending the model to include more tables allows for capturing longer-range dependencies, beyond just those between adjacent tables. For example, as shown in Figure 1, the dependency between the *Demographic* table and the *Credit Card* table can also be captured and quantified. Further details are provided in Section 5.

## 4.3 Design choices for ClavaDDPM

We detail how design decisions for ClavaDDPM meet our goals and align with our assumptions.

### 4.3.1 Relationship-aware clustering

Given the conditional independence between the parent row and its foreign key group (Equation (7)), it is important to model the latent variable $c$ such that it can effectively capture the inter-table correlation within the same foreign key group. In ClavaDDPM, we learn $c$ using Gaussian Mixture Models (GMM) in the weighted joint space of $X$ and $Y$, denoted as $H = (X; \lambda Y)$, where $\lambda$ is a weight scalar controlling the importance of child and parent tables when being clustered. Concretely, we consider $k$ clusters, and model the distribution of $h = (x; \lambda y)$ with Gaussian distributed around its corresponding centroid $c$, i.e., $P(h) = \sum_{c=1}^{k} P(c) P(h \mid c) = \sum_{c=1}^{k} \pi_c \mathcal{N}(h; \mu_c, \Sigma_c)$.

Note that diagonal GMMs are universal approximators, given enough mixtures of Gaussian distributions [49]. Therefore, we can further enforce diagonal covariance, i.e., $\Sigma_c = \text{diag}(\ldots, \sigma_l^2, \ldots)$, which, being properly optimized, immediately satisfies our assumptions that the foreign key groups are conditionally independent of their parent rows given $c$. In addition, the family of Gaussian Process Latent Variable Models (GPLVM) [30, 27, 33] has been used as an embedding technique to find low-dimensional manifolds that map to a noisy, high-dimensional space. This satisfies our need to learn a stochastic map between the noisy parent space and a condensed latent space. Thus, we can achieve a better trade-off by sacrificing some information fidelity during this quantization process while making the conditional space better shaped.

However, such clustering in the joint space $(X; \lambda Y)$ could potentially lead to inconsistency when we create the augmented table $T_Y = (Y; C)$. Though we add a weight $\lambda$ to the parent rows such that child rows with the same parent rows are likely to be assigned to the same cluster, there is still some chance that they end with different clusters. In particular, for each parent row $y_j \in Y$, its child rows are assigned to different clusters. In ClavaDDPM, we impose a majority voting step to find the most popular cluster label in each foreign key group and assign it to the parent row $y_j$. In practice, the voting agree rates tend to be high, and this can be further enforced by assigning a higher weight to the parent table (increasing $\lambda$) during GMM clustering. We evaluate the choice of $\lambda$ and voting agree rates in our ablation study in Section 5.3.

While alternative latent learning algorithms could potentially be applied, such as TabSyn [50] that demonstrated the utility of latent encoding of tabular data with VAE, this work focuses on demonstrating the effectiveness of a simple diagonal Gaussian Mixture Model (GMM) for ClavaDDPM. Our experiments (detailed in Section 5) reveal that ClavaDDPM with a diagonal GMM achieves state-of-the-art results while maintaining low computational overhead. We leave the exploration of more complex latent learning techniques for future work.

### 4.3.2 Learning with DDPM

**Gaussian diffusion backbone.** We consider one of the state-of-the-art diffusion models for single tabular data, TabDDPM [25], as the backbone model. TabDDPM models numerical data with Gaussian diffusion (DDPM [20]), and models categorical data with multinomial diffusion ([21]) with one-hot encoding, and carries out disjoint diffusion processes. However, the modeling of multinomial diffusion suffers significant performance overheads, and poses challenges to guided sampling. Instead, ClavaDDPM uses a single Gaussian diffusion backbone to model both numerical and categorical data in a unified space, where categorical data is mapped to the numerical space through label-encoding. To be specific, for a categorical feature with $m$ distinct values $C = \{c_1, \ldots, c_m\}$, a label encoding $E : C \to \{0, \ldots, m-1\}$ maps each unique category $c_i$ to an assigned unique integer value. For a table row $x = [x_{num}; \cdots; x_{cat_i}; \cdots]$, where $x_{num}$ represents all the numerical features and $x_{cat_i}$ represent a categorical feature, we obtain the unified feature by $x_{uni} = [x_{num}; \cdots; E(x_{cat_i}); \cdots]$. Based on this encoding, we learn $p_\theta(y, c)$ on the augmented parent table $T_Y = (Y; C)$ through training a Gaussian diffusion model on the unified feature space $(Y_{uni}; E(C))$.

**Classifier guided synthesis.** As defined in Equation (10), we model $p\left(x_j^i \mid c_j\right)$ by leveraging classifier-guided sampling of diffusion models, following [10]. In practice, with the sheer power of diffusion models, we jointly model $p(x \mid c)$ for the entire table without distinguishing $j$. First, we train a Gaussian diffusion model $p_\phi$ on child table row $x$, with its reverse process modeled with $x_t \sim \mathcal{N}(x_{t+1}; \mu_{\phi_{t+1}}, \Sigma_{\phi_{t+1}})$. Then, we train a classifier that classifies cluster labels based on $x$. The conditional reverse process can be approximated by $x_t \mid c \sim \mathcal{N}(x_{t+1}; \mu_{\phi_{t+1}} + \eta \Sigma_{\phi_{t+1}} g_{\psi_{t+1}}, \Sigma_{\phi_{t+1}})$, where $g_{\psi_{t+1}} = \nabla_{x_{t+1}} \log\left(p_\phi\left(c \mid x_{t+1}\right)\right)$ and $\eta$ is a scale parameter controlling the strength of conditioning. One can regard $\eta$ as a hyper parameter measuring the trade-off between single-table generation quality and inter-table correlations, to be demonstrated by our ablation study in Section 5.3.

### 4.3.3 Multi-parent dilemma: matching

Consider the case where some child table $X$ has two parent tables $Y_1, Y_2$. Our parent-child synthesis modeling paradigm would lead to two divergent synthetic child tables $\tilde{X}_1 \sim X \mid Y_1$, and $\tilde{X}_2 \sim X \mid Y_2$ Each synthetic table encodes its own parent-child relationship, i.e. the foreign keys. Combining $\tilde{X}_1$ and $\tilde{X}_2$ so that the synthetic child table contains foreign keys from both parents $p_1$ and $p_2$ is non-trivial, and we call it a multi-parent dilemma. One possible approach is to explicitly constrain the model sample space of $X \mid Y_2$ to be the synthetic data $\tilde{X}_1$, as used in PrivLava [5]. However, this approach is not applicable to diffusion models that sample from a continuous space.

We provide a ***universal*** solution for all generative models. Consider some real data point $x$ with two parent rows $y_1^j$ and $y_2^k$. Ideally, some synthetic data point $\tilde{x}$ following the same distribution as real data point $x$ should be sampled from $x \mid y_1^j, y_2^k$. This can be approximated by finding the intersection of two conditional distributions $x \mid y_1$ and $x \mid y_2$. Specifically, we estimate $\tilde{x}$ by finding two synthetic data points $\tilde{x}_1 \in \tilde{X}_1$ and $\tilde{x}_2 \in \tilde{X}_2$, such that $\tilde{x}_1 \sim x \mid y_1^j$ and $\tilde{x}_2 \sim x \mid y_2^k$, and the two points are close enough. We reason as follows: although $\tilde{x}_1$ was sampled from $x \mid y_1$, as long as it is close enough to some other synthetic data point $\tilde{x}_2$ sampled from $x \mid y_2$, then $\tilde{x}_1$ will also be within in the high density region of the distribution $x \mid y_2$, indicating a high probability that $\tilde{x}_1$ follows $x \mid y_1, y_2$. Symmetrically, the same reasoning also holds for $\tilde{x}_2$.

Therefore, we can estimate the true sample data point by $\tilde{x} = f(\tilde{x}_1, \tilde{x}_2)$ if $\tilde{x}_1$ is close to $\tilde{x}_2$, where $f$ can simply be an interpolation between two data points in practice. We call this a matching process between two divergent synthetic tables $\tilde{X}_1$ and $\tilde{X}_2$, and this can be done efficiently using approximate nearest neighbor search. Although we call this a "matching", it does not require finding a one-to-one mapping. Note that this estimate can be further improved by resampling $\tilde{X}_1$ and $\tilde{X}_2$ and estimate $\tilde{X}$ with more data points rather than just a pair, and the trade-off is a larger computational overhead, and we leave this for future research. Empirically, sampling $\tilde{X}_1$ and $\tilde{X}_2$ only once is already strong, and an ablation study on the effectiveness of parent matching is in Section 5.3.

# 5 Evaluation

We evaluate ClavaDDPM's performance in multi-relational data synthesis, using both single-table and multi-tables utility metrics (including the new long-range dependency). We present an end-to-end comparison of ClavaDDPM to the SOTA baselines, followed by an ablation study for ClavaDDPM.

## 5.1 Experimental setup

**Real-world datasets.** We experiment with five real-world multi-relational datasets including *California* [6], *Instacart 05* [23], *Berka* [4], *Movie Lens* [39, 32], and *CCS* [32]. These datasets vary in the number of tables, the maximum depth, the number of constraints, and complexity. Among all, *Berka*, *Movie Lens*, and *CCS* exhibits complex multi-parent and multi-children structures. We use *Berka* in our work for ablation study and model anatomy. Details can be found in Appendix C.1.

**Baselines.** We adopt two multi-relational synthesis models in literature as our baselines: PrivLava [5] as a representative of state-of-the-art marginal-based methods, and SDV [35] as a statistical method specially designed for multi-relational synthesis. We also introduce two multi-relational synthesis pipelines, SingleT(ST) and Denorm(D), as our additional baselines. SingleT learns and generates each table individually, but it also assigns foreign keys to each synthetic child table accordingly to the real foreign key group size distribution such that the group size information is preserved. Denorm follows the baseline idea that joins table first, but it is hard to join all tables into a single table. Hence, Denorm first applies single-table backbone model to generate the joined table between every parent-child table pair and then split it. For these two pipelines, we use CTGAN [47] and TabDDPM [25] as the single-table backbone models, representing the SOTA tabular synthesis algorithms with GAN-based models and diffusion-based models. The details can be found in Appendix C.2.

**Evaluation metrics.** We evaluate the quality of the synthetic data using: 1) *cardinality* to measure the foreign key group size distribution for the intra-group correlations; 2) *column-wise density estimation (1-way)* to estimate the density of every single column for all tables; 3) *pair-wise column correlation (k-hop)* for the correlations of columns from tables at distance $k$, e.g., 0-hop refers to columns within the same table and 1-hop refers to a column and another column from its parent or child table; 4) *average 2-way*, which computes the average of all $k$-hop column-pair correlations, taking into consideration of both short-range ($k = 0$) and longer-range ($k > 0$) dependencies. For each measure, we report the complement of Kolmogorov-Smirnov (KS) statistic and total variation (TV) distance [3] between the real data and the synthetic data, ranging from 0 (the worst utility) to 1 (the best utility). The reported results are averaged over 3 randomly sampled synthetic data.

We also consider higher-order single-table evaluation metrics for some representative tables as prior work [50]. We include their details and experiemntal results in Appendix D due to space constraints.

All experiments are conducted with an NVIDIA A6000 GPU and 32 CPU cores, with a time limit of 7 days. If an algorithm fails to complete within the time limit, we report TLE (time limit exceeded). Implementation details and hyperparameter specifics are in Appendix C.3.

## 5.2 End-to-end evaluation

We conducted multi-table synthesis experiments on five multi-table datasets and report the averaged utility with standard deviation for all algorithms in Table 1. First, the evaluation shows that ClavaD-DPM has an overall advantage against all the baseline models in terms of correlation modeling, and is surpassing the baselines by larger margins for longer-range dependencies. e.g. in *Instacart 05*, our model outperforms the best baseline by $58.29\%$ on 2-hop correlations, and in *Berka*, our model exceeds the best baseline by $20.24\%$ on 3-hop correlations. For single-column densities and cardinality distributions, ClavaDDPM exhibits a competitive result compared to the state-of-the-art baseline models. We also evaluate ClavaDDPM against baselines on high-order single-table metrics (Appendix D.3), which shows that our model has advantages in preserving data fidelity, generating diverse data, and achieving high machine learning efficacy.

It is worth noting that ClavaDDPM, despite its complexity and capability, is more efficient and robust than some simpler baselines. PrivLava demonstrates strong performance on the California dataset

---

[3]The complement to KS/TV distance between two distributions $P$ and $Q$ is $1.0 - D_{\text{KS/TV}}(P||Q)$. We use KS for numerical values and TV for categorical values.

| End-to-end | PrivLava | SDV | ST-CTGAN | ST-TabDDPM | ST-ClavaDDPM | D-CTGAN | D-TabDDPM | D-ClavaDDPM | ClavaDDPM |
|---|---|---|---|---|---|---|---|---|---|
| **California** | | | | | | | | | |
| CARDINALITY | 99.90 ±0.03 | 71.45 ±0.00 | 99.93 ±0.02 | 99.94 ±0.00 | 99.89 ±0.04 | 99.90 ±0.07 | 99.94 ±0.00 | 99.87 ±0.02 | 99.19 ±0.29 |
| 1-WAY | 99.71 ±0.02 | 72.32 ±0.00 | 91.59 ±0.50 | 83.27 ±0.07 | 99.51 ±0.04 | 91.22 ±0.07 | 93.10 ±0.84 | 94.99 ±0.02 | 98.77 ±0.02 |
| 0-HOP | 98.49 ±0.05 | 50.23 ±0.00 | 87.67 ±0.63 | 79.27 ±0.08 | 98.69 ±0.08 | 86.58 ±0.44 | 91.12 ±1.35 | 94.17 ±0.01 | 97.65 ±0.05 |
| 1-HOP | 97.46 ±0.12 | 54.89 ±0.00 | 84.82 ±0.61 | 78.44 ±0.04 | 92.96 ±0.05 | 82.72 ±0.30 | 84.43 ±1.80 | 87.24 ±0.10 | 95.16 ±0.39 |
| AVG 2-WAY | 97.97 ±0.09 | 52.56 ±0.00 | 86.25 ±0.60 | 78.85 ±0.06 | 95.83 ±0.07 | 84.65 ±0.35 | 87.78 ±1.57 | 90.71 ±0.04 | 96.41 ±0.20 |
| **Instacart 05** | | | | | | | | | |
| CARDINALITY | | | 95.78 ±0.96 | | 94.73 ±0.14 | 93.81 ±0.39 | | 94.98 ±0.84 | 95.30 ±0.79 |
| 1-WAY | | | 79.85 ±0.96 | | 89.30 ±0.00 | 69.07 ±0.57 | | 71.83 ±0.32 | 89.84 ±0.29 |
| 0-HOP | DNC | DNC | 78.27 ±0.28 | TLE | 99.70 ±0.00 | 84.85 ±0.44 | TLE | 88.74 ±0.00 | 99.62 ±0.04 |
| 1-HOP | | | 62.48 ±0.16 | | 66.93 ±0.07 | 60.26 ±0.38 | | 62.58 ±0.05 | 76.42 ±0.39 |
| 2-HOP | | | 24.82 ±8.02 | | 16.22 ±13.41 | 0.00 ±0.00 | | 0.00 ±0.00 | 39.29 ±3.38 |
| AVG 2-WAY | | | 60.05 ±1.40 | | 66.66 ±2.37 | 56.19 ±0.33 | | 58.52 ±0.03 | 76.02 ±0.78 |
| **Berka** | | | | | | | | | |
| CARDINALITY | | | 96.08 ±0.18 | 68.29 ±0.00 | 97.06 ±0.80 | 97.72 ±0.29 | 97.71 ±0.00 | 96.06 ±1.15 | 96.92 ±0.71 |
| 1-WAY | | | 79.78 ±0.75 | 76.41 ±2.21 | 94.58 ±0.01 | 83.00 ±0.65 | 80.09 ±0.68 | 83.28 ±0.97 | 94.29 ±0.44 |
| 0-HOP | | | 74.24 ±0.32 | 72.80 ±1.23 | 91.72 ±0.23 | 76.04 ±0.34 | 74.82 ±0.49 | 72.12 ±0.73 | 91.49 ±0.82 |
| 1-HOP | DNC | DNC | 66.59 ±0.54 | 54.01 ±2.35 | 81.77 ±1.19 | 75.25 ±0.55 | 61.99 ±2.10 | 55.77 ±2.80 | 86.86 ±2.74 |
| 2-HOP | | | 75.83 ±1.07 | 59.88 ±1.39 | 78.09 ±0.53 | 72.40 ±0.43 | 63.94 ±1.33 | 57.68 ±1.67 | 89.25 ±2.27 |
| 3-HOP | | | 72.58 ±0.86 | 55.29 ±1.58 | 75.56 ±0.34 | 71.74 ±0.69 | 62.67 ±2.26 | 55.59 ±1.48 | 87.27 ±1.92 |
| AVG 2-WAY | | | 73.22 ±0.45 | 61.74 ±1.57 | 82.33 ±0.40 | 73.94 ±0.37 | 66.29 ±1.30 | 60.93 ±1.49 | 89.21 ±1.95 |
| **Movie Lens** | | | | | | | | | |
| CARDINALITY | | | 98.91 ±0.06 | | 98.99 ±0.16 | 98.70 ±0.40 | | 98.87 ±0.26 | 99.07 ±0.18 |
| 1-WAY | | | 86.58 ±0.80 | | 99.19 ±0.00 | 68.38 ±0.36 | | 78.03 ±0.17 | 99.34 ±0.10 |
| 0-HOP | DNC | DNC | 72.80 ±0.86 | TLE | 98.56 ±0.01 | 31.96 ±0.32 | TLE | 57.33 ±0.10 | 98.69 ±0.15 |
| 1-HOP | | | 74.86 ±0.63 | | 92.72 ±0.09 | 58.00 ±0.05 | | 77.45 ±1.93 | 96.19 ±0.11 |
| AVG 2-WAY | | | 74.10 ±0.62 | | 94.87 ±0.06 | 48.45 ±0.09 | | 70.07 ±1.19 | 97.11 ±0.02 |
| **CCS** | | | | | | | | | |
| CARDINALITY | | 74.36 ±8.40 | 99.00 ±0.53 | 93.70 ±0.00 | 99.37 ±0.16 | 26.98 ±0.05 | 26.97 ±0.00 | 26.70 ±0.20 | 99.25 ±0.16 |
| 1-WAY | | 69.04 ±4.38 | 82.21 ±0.32 | 82.72 ±0.06 | 95.20 ±0.40 | 73.68 ±0.35 | 79.28 ±0.10 | 79.29 ±0.13 | 92.37 ±2.30 |
| 0-HOP | DNC | 94.84 ±1.00 | 87.02 ±0.18 | 88.10 ±0.07 | 98.96 ±0.00 | 81.70 ±0.33 | 87.15 ±0.16 | 86.60 ±0.14 | 98.47 ±0.79 |
| 1-HOP | | 21.74 ±9.62 | 49.84 ±2.30 | 47.11 ±0.06 | 51.62 ±0.22 | 56.86 ±0.66 | 61.53 ±1.50 | 57.77 ±0.69 | 83.15 ±4.22 |
| AVG 2-WAY | | 41.68 ±6.73 | 59.98 ±1.72 | 58.29 ±0.06 | 64.53 ±0.16 | 63.64 ±0.57 | 68.51 ±1.11 | 65.64 ±0.50 | 87.33 ±3.12 |

Table 1: End-to-end results. DNC denotes *Did Not Converge*, and TLE denotes *Time Limit Exceeded*. ST stands for Single-T and D stands for Denorm. Statistical metrics described in Section 5.1 are reported.

| | Default | Varying $k$ | | Varying $\lambda$ | | | Varying $\eta$ | | |
|---|---|---|---|---|---|---|---|---|---|
| **Berka** | $k=20, \lambda=1.5, \eta=1$ | $k=1$ | $k=1000$ | $\lambda=0$ | $\lambda=10$ | $\lambda=100$ | $\eta=0$ | $\eta=2$ | No Matching |
| CARDINALITY | 96.92 ±0.71 | 97.05 ±0.40 | 95.12 ±0.85 | 97.21 ±0.40 | 97.22 ±0.39 | 97.17 ±0.47 | 96.89 ±0.24 | 96.95 ±0.29 | 97.76 ±0.36 |
| 1-WAY | 94.29 ±0.44 | 94.04 ±0.60 | 93.73 ±0.55 | 94.30 ±0.57 | 94.64 ±0.45 | 94.82 ±0.44 | 94.67 ±0.39 | 94.14 ±0.49 | 94.71 ±0.29 |
| 0-HOP | 91.49 ±0.82 | 87.96 ±2.02 | 89.67 ±0.27 | 88.60 ±2.11 | 89.94 ±0.80 | 90.84 ±1.37 | 88.55 ±1.20 | 90.40 ±0.52 | 88.75 ±1.02 |
| 1-HOP | 86.86 ±2.74 | 77.31 ±0.85 | 84.97 ±1.33 | 81.72 ±5.11 | 84.62 ±1.65 | 84.19 ±1.88 | 81.63 ±1.23 | 85.19 ±1.84 | 81.97 ±2.00 |
| 2-HOP | 89.25 ±2.27 | 80.78 ±0.68 | 88.18 ±1.09 | 83.24 ±4.35 | 87.78 ±1.54 | 85.64 ±2.52 | 84.42 ±0.43 | 87.64 ±1.14 | 82.41 ±1.70 |
| 3-HOP | 87.27 ±1.92 | 75.53 ±2.75 | 86.10 ±1.63 | 77.15 ±7.05 | 85.29 ±2.16 | 79.19 ±5.35 | 80.66 ±3.16 | 82.58 ±3.76 | 74.78 ±2.00 |
| AVG 2-WAY | 89.21 ±1.95 | 81.64 ±1.09 | 87.77 ±0.80 | 84.01 ±3.98 | 87.52 ±1.36 | 86.36 ±2.12 | 84.69 ±0.26 | 87.57 ±0.89 | 83.59 ±1.59 |
| AVG AGREE-RATE | 81.12 ±0.99 | 100.00 ±0.00 | 65.97 ±0.17 | 78.85 ±0.73 | 80.87 ±0.73 | 82.44 ±0.77 | 81.32 ±0.80 | 81.37 ±1.09 | 80.88 ±0.58 |

Table 2: Ablation studies on number of clusters $k$, parent scale $\lambda$, and classifier gradient scale $\eta$. Note that $\eta$ and matching have no effect on agree rates. Statistical metrics described in Section 5.1 are reported.

(the simplest data), but fails to converge on all the other datasets. SDV also tends to fail on complex datasets, and is limited to datasets with at most 5 tables and maximum depth of 2 [9]. Although TabDDPM shares a similar model backbone with ClavaDDPM, its synthesis fails to complete within 7 days on multiple datasets, while ClavaDDPM completes all experiments within 2 days.

## 5.3 Ablation study

**Gaussian diffusion backbone.** To decouple the effect of our Gaussian diffusion only backbone with the latent conditioning training paradigm, we also included two models in Table 1: ST-ClavaDDPM and D-ClavaDDPM, which use the Gaussian diffusion model in ClavaDDPM as backbone model, but are trained and synthesized following Single-T and Denorm. Compared to other baselines, ST-ClavaDDPM exhibits superiority in modeling both single column densities and column-pair correlations. ST-ClavaDDPM significantly outperforms its sibling ST-TabDDPM, which proves the effectiveness of using Guassian diffusion for tabular data synthesis solely. On the other hand, ST-ClavaDDPM falls short on longer-range correlations when compared to the full ClavaDDPM model. This observation provides solid evidence to the efficacy of our multi-table training paradigm.

Besides the study of the single-table backbone models, we perform a comprehensive ablation study using *Berka* (for it has the most complex multi-table structure) on each component of ClavaDDPM and provide empirical tuning suggestions. The full results are in Table 2.

**Number of clusters** $k$**.** We study the necessity of using latent cluster conditioning: (i) no conditioning with $k = 1$; (ii) many clusters with $k = 1000$ to approximate a direct conditioning on parent rows rather than latent variables. When $k = 1$, the quality of long-range correlation degrades drastically. When $k = 1000$, we still get reasonably strong performance, which showcases ClavaDDPM's robustness. Compared to the default setting ($k = 20$), the metrics are lower in all of cardinality distribution, single column densities, and column correlations — proper latent variable learning leads to better results than direct conditioning on parent rows. We also report a new metric *avg agree-rate*, the average of all per-table agree rates for the labels within each foreign key group (Section 4.3.1). This measure highly depends on $k$, but a higher rate does not always imply a better performance (e.g., k=1 achieves perfect rates). We provide more insights on how it varies with the next parameter. We also conducted finer-grained experiments to examine the effect of $k$ on model performance, as shown in Appendix D.2.

**Parent scale** $\lambda$**.** Varying the parent scale parameter $\lambda$ changes the agree-rates as shown in Table 2, but the downstream model performance does not vary too much. This result indicates that the relation-aware clustering process is robust against such factors, and the GMM model is capable of capturing nuances in data patterns. The detailed discussion is in Appendix D.1.

**Classifier gradient scale** $\eta$**.** This parameter controls the magnitude of classifier gradients when performing guided sampling, and thus the trade-off between the sample quality and conditional sampling accuracy. Table 2 shows that, when $\eta = 0$, which essentially disables classifier conditioning, the single column densities (1-way) are slightly higher than the default setting. However, it falls short in capturing long-range correlations. When $\eta = 2$, the conditioning is emphasized with a higher weight, which significantly improves the modeling of multi-hop correlations compared to $\eta = 0$ case.

**Comparing with no matching for multi-parent dilemma.** *Berka* (Figure 1) suffers from the multi-parent dilemma , where the *Disposition* table has two parent tables, *Account* and *Client*. Our abalation study switch the table matching technique to a naive merging of two synthetic table (Appendix C.2). The experiment result show that even if trained with the same hyper parameters and model structures, ClavaDDPM with matching is significantly stronger than the no-matching setup in terms of long-range correlations, with 3-hop correlations $16.70\%$ higher than no-matching.

## 6 Conclusion

We proposed ClavaDDPM as a solution to the intricate problem of synthetic data generation for multi-relational data. ClavaDDPM utilizes clustering on a child table to learn the latent variable that connects the table to its parents, then feeding them to the diffusion models to synthesis the tables. We presented ClavaDDPM's seamless extension to multiple parents and children cases, and established a comprehensive multi-relational benchmark for a through evaluation – introducing a new holistic multi-table metric *long-range dependency*. We demonstrated ClavaDDPM not only competes closely with the existing work on single-table synthesis metrics, but also it outperforms them in ranged (inter-table) dependencies. We deliberately selected the more complex public databases to exhibit ClavaDDPM's scalability, and introduce it as a confident candidate for a broader impact in industry.

We focused on foreign key constraints in this work, and made the assumption that child rows are conditionally independent given corresponding parent rows. This brings three natural follow-up research directions: i) extension to the scenarios where this prior information is not available and these relationships need to be discovered first[29], ii) further relaxing the assumptions, and iii) inspecting multi-relational data synthesis with other integrity constraints (e.g, denial constraints[15], general assertions for business rules). Furthermore, we evaluated ClavaDDPM's privacy with the common (in tabular data literature) DCR metric. Nonetheless, we think it is worthwhile to: i) evaluate the resiliency of ClavaDDPM against stronger privacy attacks[43], and ii) investigate the efficacy of boosting ClavaDDPM with privacy guarantees such as differential privacy. Similarly, the impacts of our design on fairness and bias removal, as another motivating pillar in synthetic data generation, is well worth exploring as future work. We believe the thorough multi-relational modeling formulation we presented in this work, can serve as a strong foundation to build private and fair solutions upon.

## Acknowledgments

This work was supported by NSERC through a Discovery Grant, an alliance grant, the Canada CIFAR AI Chairs program. Resources used in preparing this research were provided, in part, by the Province of Ontario, the Government of Canada through CIFAR, and companies sponsoring the Vector Institute. We thank the reviewers and program chairs for their detailed comments, which greatly improved our paper.

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

## A   Notation Summary

We use boldface to represent random variables. e.g. $Y \sim \boldsymbol{Y}$, where $Y$ is the data of $R_2$, and $\boldsymbol{Y}$ is the random variable $\boldsymbol{Y}$ being sampled from. In addition, we use subscript to represent the **parent row** some data or random variable refers to. e.g., $\boldsymbol{x}_j$ represents the child random variable that refers to parent $\boldsymbol{y}_j$. The important notations used in the paper are summarized in Table 3.

| | |
|---|---|
| Relational database, relational table, synthetic table | $\mathcal{R}, R, \tilde{R}$ |
| Random variable and data for a parent table | $\boldsymbol{Y}, Y$ |
| Random variable and data for a child table | $\boldsymbol{X}, X$ |
| Random variable and data for a grandchild table | $\boldsymbol{Z}, Z$ |
| Random variable and data for $j$th parent row | $\boldsymbol{y}_j, y_j$ |
| Random variable and data for foreign key group referring to $j$th parent row | $\boldsymbol{g}_j, g_j$ |
| Random variable and data for child rows in $j$th foreign key group | $\boldsymbol{x}_j^i, x_j^i$ |
| Random variable and data for $j$th foreign key group size | $\boldsymbol{s}_j, s_j$ |
| Latent cluster random variable and value | $\boldsymbol{c}, c$ |
| Augmented table with a latent variable column | $T_Y = (Y; C)$ |
| Directed acyclic graph, nodes, edges | $\mathcal{G} = (\mathcal{R}, \mathcal{E})$ |
| Diffusion model for augmented table $T$ | $p_T$ |
| Diffusion model for child data paremeterized by $\phi$ | $p_\phi(x).$ |
| Latent variable classifier parameterized by $\psi$ | $p_\psi(c \mid x)$ |
| Classifier guided distribution, parameterized by $\phi, \psi$ | $p_{\phi,\psi}(x \mid c)$ |

Table 3: Notation summary

## B   Algorithm Details

### B.1   Diagram for two-table relational databases

Figure 2 summarizes the generative process for two-table cases.

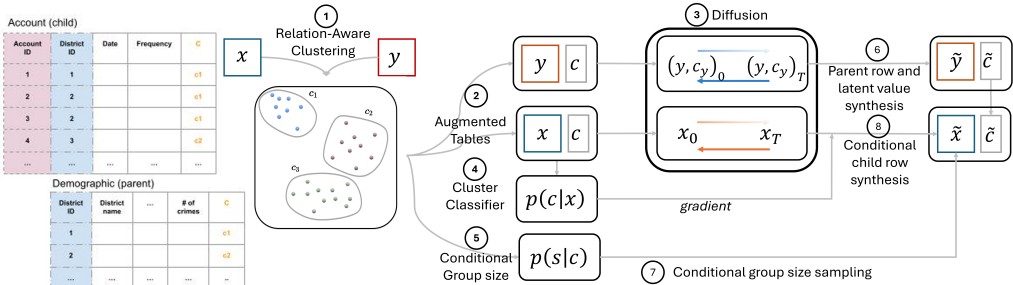

Figure 2: ClavaDDPM overview for a two-table relational database

### B.2   End-to-end algorithms for more tables

We detail the end-to-end algorithms for the three phases of ClavaDDPM, including (i) latent learning and table augmentation, (ii) training, and (iii) synthesis.

**Latent learning and table augmentation.**   As shown in Algorithm 1, given a database $\mathcal{R} = \{R_1, \ldots, R_m\}$ and foreign key constraint graph $\mathcal{G}$, we learn the set of latent variables $C_{i,j}$ for every pair of parent-child $(R_i \rightarrow R_j) \in \mathcal{G}.\mathcal{E}$ and augment all the latent variables to the parent table and the child table, denoted by $T_j$ and $T_i'$, respectively. We initialize each augment table with its original table (line 1). This algorithm follows a bottom-up topological order starting from the leaf child with its parent (line 2), ensuring each child table is already augmented by the time we learn the latent variable to augment its parent. For each parent-child pair $R_i \rightarrow R_j$, we join $T_i$ (not $T_i'$) with $R_j$ into

a single table $(X;Y)$ (line 3) and then run the clustering algorithm using GMM and maximum voting described in Section 4.3.1. We append the corresponding clustering labels $C_{i,j}$ to the augmented parent table $T_j$ and augmented child table $T_i$, respectively.

---

**Algorithm 1** ClavaDDPM: Latent learning and table augmentation.

---

    **Input:** tables $\mathcal{R} = \{R_1, \ldots, R_m\}$, foreign key constraint graph $\mathcal{G}$
    **Output:** latent variables $\{C_{i,j} | (R_i \rightarrow R_j) \in \mathcal{G}.\mathcal{E}\}$, augmented parent tables $\{T_1, \ldots, T_m\}$,
augmented child tables $\{T'_1, \ldots, T'_m\}$
  1: Initialize augmented tables $\{T_1, \ldots, T_m\} \leftarrow \mathcal{R}$, $\{T'_1, \ldots, T'_m\} \leftarrow \mathcal{R}$
  2: **for** $(R_i \rightarrow R_j)$ in bottom-up topological order of $\mathcal{G}$ **do**
  3:     Join parent and augmented child $(X;Y) \leftarrow (T_i, R_j)$
  4:     $C_{i,j} \leftarrow Clustering\,(X;Y)$         ▷ Relationship-aware clustering in Section 4.3.1
  5:     Augment parent $T_j \leftarrow (T_j; C_{i,j})$
  6:     Augment child $T'_i \leftarrow (T'_i; C_{i,j})$.
  7: **end for**

---

**Training.** As shown in Algorithm 2, the training phase takes in the augmented parent tables $\{T_1, \ldots, T_m\}$ and the foreign key constraint graph $\mathcal{G}$. For each augmented table $T_j$, we train a diffusion model $p_{T_j}$ (lines 2-4). Then, for each parent-child pair $R_i \rightarrow R_j$ (lines 5-7), we train a child classifier $p_\phi(c_{i,j}|x)$ with $R_i$'s child augment table $T'_i$, where the latent column $C_{i,j}$ is used as labels, and all remaining columns are used as training data (including the augmented latent columns corresponding to $R_i$'s children). Using the same table, we also estimate the foreign key group size distribution conditioned on the latent variable $p(s|c_{i,j})$.

---

**Algorithm 2** ClavaDDPM: Training

---

    **Input:** augmented parent tables $\{T_1, \ldots, T_m\}$, augmented child tables $\{T'_1, \ldots, T'_m\}$, foreign key constraint graph $\mathcal{G}$
    **Output:** diffusion models $\mathcal{D}$, classifiers $\mathcal{C}$, group size distributions $\mathcal{S}$
  1: Initialize $\mathcal{D}, \mathcal{C}, \mathcal{S} \leftarrow \emptyset$
  2: **for** $R_j$ in $\mathcal{G}.\mathcal{R}$ **do**
  3:     Train $p_{T_j}$ with $T_j$, and add to $\mathcal{D}$
  4: **end for**
  5: **for** $(R_i \rightarrow R_j)$ in topological order of $\mathcal{G}$ **do**
  6:     Learn classifier $p_\phi(c_{i,j}|x)$ and $p(s|c_{i,j})$ using with $T'_i$ (ignoring irrelevant latent columns) and add to $\mathcal{C}$ and $\mathcal{S}$ respectively
  7: **end for**

---

**Synthesis.** Algorithm 3 takes in learned diffusion models $\mathcal{D}$, classifiers $\mathcal{C}$, group size distributions $\mathcal{S}$, and the DAG representation of the database $\mathcal{G}$, and outputs the synthetic database $\tilde{\mathcal{R}} = \left\{\tilde{R}_1, \ldots, \tilde{R}_m\right\}$. We first initialize the synthetic augmented tables to be empty (line 1). Then, for root augmented tables, since they have no parents to condition on, they can be directly synthesized from their diffusion models (line 2-4). Next, we traverse the database in topological order to synthesize the remaining augmented tables (line 5-16): If we have already synthesized $\tilde{T}_i$ before, which means we encounter the multi-parent dilemma, we just store the old version and continue to generate a new version (line 6-9). For each parent-child relationship $R_i \rightarrow R_j$, we must have already sampled the augmented parent table $\tilde{T}_j$. This is because we follow the topological order of a DAG, and all root augmented tables have been synthesized as base cases. Therefore, we can obtain the synthetic latent variables $\tilde{C}_{i,j}$ from the synthetic augmented parent $\tilde{T}_j$ (line 10). Then, we iterate through each synthetic latent value $\tilde{c}_{i,j}$ and perform a two-step sampling: (1) use the learned group size distribution to conditionally sample a group size $\tilde{s}$ (line 12); (2) sample $\tilde{s}$ rows of data conditioned on $\tilde{c}_{i,j}$ using classifier guided sampling (line 13). We repeat this process until the augmented child table $\tilde{T}_i$ is fully synthesized. We simply obtain synthetic tables from synthetic augmented tables by removing all synthetic latent columns (line 17-19). Finally, for all the encountered multi-parent dilemmas, we follow Section 4.3.3 to match the divergent versions.

**Algorithm 3** ClavaDDPM: Synthesis

    **Input:** diffusion models $\mathcal{D}$, classifiers $\mathcal{C}$, group size distributions $\mathcal{S}$, foreign key constraint graph $\mathcal{G}$

    **Output:** Synthetic tables $\left\{ \tilde{R}_1, \ldots, \tilde{R}_m \right\}$

1: Initialize $\tilde{T}_1, \ldots, \tilde{T}_m \leftarrow \emptyset, \ldots, \emptyset$
2: **for** $R_j$ in root nodes **do**
3:      Sample $\tilde{T}_j \sim p_{T_j}$
4: **end for**
5: **for** $(R_i \rightarrow R_j)$ in topological order of $\mathcal{G}$ **do**
6:      **if** $\tilde{T}_i$ already synthesized **then**
7:          Store $\tilde{T}_i$ as $\tilde{T}_{i,k}$, where $k$ is the parent that synthesized $\tilde{T}_{i,k}$
8:          Reinitialize $\tilde{T}_i \leftarrow \emptyset$
9:      **end if**
10:      Split $\tilde{T}_j$ into $\left( \cdot \; ; \tilde{C}_{i,j} \right)$
11:      **for** $\tilde{c}_{i,j}$ in $\tilde{C}_{i,j}$ **do**
12:          Sample $\tilde{s} \sim p\left( s \mid \tilde{c} \right)$
13:          Classifier-guided sample $\tilde{s}$ rows of data: $\tilde{t}_i \sim p\left( t_i \mid \tilde{c}_{i,j} \right)$
14:          Append $\tilde{t}_i$ to $\tilde{T}_i$
15:      **end for**
16: **end for**
17: **for** $\tilde{T}_j$ in all synthetic augmented tables **do**
18:      $\tilde{R}_j \leftarrow$ all latent columns removed from $\tilde{T}_j$
19: **end for**
20: **for** $\tilde{R}_j$ with multiple synthetic versions **do**
21:      $\tilde{R}_j \leftarrow \text{MATCHING}\left( \tilde{R}_{j,p_1}, \ldots, \tilde{R}_{j,p_q} \right)$
22: **end for**

| | # Tables | # Foreign Key Pairs | Depth | Total # Attributes | # Rows in Largest Table |
|---|---|---|---|---|---|
| *California* | 2 | 1 | 2 | 15 | $1,690,642$ |
| *Intacart 05* | 6 | 6 | 3 | 12 | $1,616,315$ |
| *Berka* | 8 | 8 | 4 | 41 | $1,056,320$ |
| *Movie Lens* | 7 | 6 | 2 | 14 | $996,159$ |
| *CCS* | 5 | 4 | 2 | 11 | $383,282$ |

Table 4: Dataset Specifics

## C   Experimental Details

### C.1   Datasets

Here we describe the real-world datasets used in our evaluation in detail. The specifics of datasets are in Table 4.

**California**: The California dataset is a real-world census database ([6]) on household information. It consists of two tables in the form of a basic parent-child relationship.

**Instacart 05**: The Instacart 05 is created by downsampling 5-percent from the Kaggle competition dataset Instacart ([23]), which is a real-world transaction dataset of instacart orders. This dataset consists of 6 tables in total with a maximum depth of 3.

**Berka**: The Berka dataset is a real-world financial transaction dataset ([4]), consisting of 8 tables with a maximum depth of 4. This will be the main dataset in our work for ablation study and model anatomy.

**Movie Lens**: The Movie Lens dataset ([39], [32]) consists of 7 tables with a maximum depth of 2. This dataset exhibits complex multi-parent and multi-children structures.

**CCS**: The CCS dataset ([32]) is a real-world transactional dataset Czech debit card company. It consists of 5 tables with a maximum depth of 2, which exhibits complex multi-parent and multi-children patterns.

## C.2   Baselines

We adopt two multi-relational synthesis models in literature as our baselines: PrivLava [5] as a representative of state-of-the-art marginal-based methods, and SDV [35] as a statistical method specially designed for multi-relational synthesis. In addition, we introduce two types of multi-relational synthesis pipelines, SingleT and Denorm, as our additional baselines. For the additional baselines, we use CTGAN ([47]) and TabDDPM ([25]) as backbone models, representing the state-of-the-art tabular synthesis algorithms with GAN-based models and diffusion-based models. In the following, we describe the high-level ideas of Single-T and Denorm.

**Single-T**: Given a single-table backbone model, we first learn and synthesize each table individually. Then, for each parent-child table pair $(p, c)$, we assign foreign keys to the synthetic child table $\tilde{R}_c$ by randomly sampling group sizes in the real table $R_c$, which enforces the synthetic group size distributions to be similar to real ones.

**Denorm**: For each parent-child table pair $(p, c)$, we join the table into $R_{p,c}$, then use the single-table backbone model to synthesize the joint table $\tilde{R}_{p,c}$. Finally, we split $\tilde{R}_{p,c}$ into two synthetic tables $\tilde{R}_p$ and $\tilde{R}_c$ as follows: (1) Lexicographically sort $\tilde{R}_{p,c}$, where the parent columns are prioritized. This guarantees that similar parent records are grouped together. (2) From the real table $R_c$, randomly sample group sizes $\tilde{g}$ with replacement. Then, for each sampled $\tilde{g}$, the consecutive $\tilde{g}$ rows in $\tilde{R}_{p,c}$ will be taken as a synthetic foreign key group $\tilde{g}_{p,c}$. The child columns part of $\tilde{g}_{p,c}$ will be assigned the same foreign key and appended to the child synthetic table $\tilde{R}_c$. Then, we randomly sample a parent row in $\tilde{g}_{p,c}$ and append to the parent synthetic table $\tilde{R}_p$. We follow the exact same way as in ClavaDDPM to extend 2-table Denorm to the entire database.

**Random matching:** We conduct ablation study by training a ClavaDDPM model with the same setup as the default setting, while instead of performing table matching to handle the multi-table dilemma, it performs a naive merging of two synthetic tables. For the diverged synthetic tables $\tilde{R}_{D,A}$ and $\tilde{R}_{D,C}$, where $\tilde{R}_{D,A}$ is the *Disposition* table synthesized conditioned on the *Account* table, and $\tilde{R}_{D,C}$ is conditioned on the *Client* table, we simply keep $\tilde{R}_{D,A}$, and randomly assign the $(D, C)$ foreign keys from $\tilde{R}_{D,C}$ to $\tilde{R}_{D,A}$.

## C.3   Implementation Details

### C.3.1   Classifier Training

We use an MLP for classifier with layers $128, 256, 512, 1024, 512, 256, 128$. The output layer size is adapted to the number of clusters $k$. We use learning rate of $1e - 4$, and optimize with *AdamW* optimizer, and use cross entropy loss as objective. The overall training paradigm follows [10], where we incorporate timestep information by encoding the timesteps into sinusoidal embeddings, which are then added to the data. For experiments on *California*, we train the classifier for 10000 iterations, and for all other datasets we train 20000 iterations.

### C.3.2   Hyper Parameters

**Baseline models.**   PrivLava was run under a non-private setup by setting privacy budget $\epsilon = 50$, and the datasets are prepossessed spesifically for PrivLava to have domain sizes less than 200.

For all models with ClavaDDPM or TabDDPM backbones, we use the same set of hyper parameters. We set diffusion timesteps to 2000, and use learning rate of $6e - 4$. In terms of model architecture, we use MLP with layer sizes $512, 1024, 1024, 1024, 1024, 512$. The model architecture details are following the implementation of TabDDPM [25]. All DDPM-based models are trained $100, 000$ iterations on *California* dataset, and $200, 000$ on other datasets.

|  | CALIFORNIA | INSTACART 05 | BERKA | MOVIE LENS | CCS |
|---|---|---|---|---|---|
| Num Clusters $k$ | 25 | 50 | 20 | 50 | 25 |
| Parent Scale $\lambda$ | 1 | 1 | 1.5 | 1 | 1 |
| Classifier Scale $\eta$ | 1 | 1 | 1 | 1 | 1 |

Table 5: Hyper parameters of ClavaDDPM on each dataset.

We conducted CTGAN experiments using the interface from SDV library, and follows the default parameters, where the learning rates for the generator and discriminator are both $2e-4$, and is trained 300 epochs.

PrivLava's code is not publicly available, and we directly followed the authors' settings. Note that PrivLava requires a privacy budget searching process, and $\epsilon = 50$ is the largest working privacy budget according to our experiments, where larger $\epsilon$ leads to failure. We consider this as large enough to resemble a non-private setting.

For SDV, we used the default setting of their HMASynthesizer, which by default uses a Gaussian Copula synthesizer.

**ClavaDDPM settings.** We list the major hyper parameters used by ClavaDDPM for each dataset in Table 5, and we provide an empirical guidance for hyper parameter tuning: it is suggested to use number of clusters $k$ to be at least 20, and classifier scale $\eta$ to be in $[0.5, 2]$. We empirically find ClavaDDPM consistently perform well in such a range across all datasets. Parent scale $\lambda$ is a less sensitive factor, and $\lambda = 1$ is a stable starting point for tuning. In general, ClavaDDPM is robust, with a small hyper parameter space, and there is very little need for tuning.

### C.3.3 Metrics

**C2ST.** The Classifier Two Sample Test trains a logistic regression classifier to distinguish synthetic data from real data. We consider this metric as a high-level reflection of data fidelity.

**Machine Learning Efficacy.** Different from prior works that evaluate MLE utilities [25, 52, 50], who work on datasets with predefined machine learning tasks, the five real-world multi-relational datasets we use do not come with a designated downstream task. In addition, the prior knowledge about which column will be used for downstream predictions will introduce significant inductive bias to the training process, especially for models capable of performing task-oriented training. To avoid such issue, we evaluate machine learning efficacy on each of the columns. To be specific, each time we select a column as target, and train an XGBoost [8] model on remaining columns. For categorical target columns, we perform regression and evaluate $R^2$, and for categorical target columns we perform classification and evaluate $F_1$. The overall MLE is measured by *average $R^2$* and *average $F_1$* across all columns.

To evaluate the single-table MLE on synthetic data generated from multi-table synthesis process, instead of performing an independent train-test split on each table, we split by foreign key relationship. e.g. for *California* dataset, we first perform a random $90\%, 10\%$ split on the parent table *Household*, and then we follow the foreign key constraints to assign corresponding child rows, i.e. *Individual*s to the corresponding buckets. Note that although this splitting method does not lead to the same train/test ratio on child table, we consider such sampling to be foreign key relationship preserving, which is a more important property in the context of multi-table synthesis.

## D Additional Experiments

### D.1 Agree rate discussion

As introduced in Section 4.3.1, within each foreign key group, we perform a majority voting to synchronize the assigned cluster label among the group. To measure the consistency of such majority voting process, we introduce the measurement of *agree rate*, which computes the average ratio of agreeing on the mode within each foreign key group, and the metric *avg agree-rate* is the average of

all per-table agree rates within a multi-table dataset.

$$A(g) = \frac{m_g}{|g|} \tag{11}$$

$A(g)$ represents the *agree rate* of some group $g$, where $m_g$ represents the number of records that are assigned the mode cluster within $g$, and the *avg agree-rate*

$$A_{\text{AVG}} = \frac{1}{|G|} \sum_{g \in G} A(g) \tag{12}$$

is computed as the average of the agree rates across all groups.

However, our experiment results in Table 2 indicate that the relationship-aware clustering process is robust against such factors, and the GMM model is capable of capturing nuances in data patterns. In our experiments on *Berka* dataset, ClavaDDPM's clustering process achieves a consistent agree rate around $81\%$, which is practically high enough given we have 20 clusters. Intuitively, when parent scale approaches infinity, the clustering is performed completely on parent table, which will lead to a perfect agree rate. Also note that a higher agree rate does not always imply a better performance, and the disagreement can potentially come from the intrinsic parent-child relationships. e.g. when child data is intrinsically independent of parent data, it is reasonable to have noisy learned latent variables, leading to low agree rates. However, in such cases the noisy latent variable would not degenerate model performance, because the best strategy will be a direct sampling of child table, rather than conditioning on some enforced prior distribution. In addition, as shown in Table 2, agree rates are highly affected by the number of clusters chosen, and there exists a trade-off between the granularity in clustering and consistency. In an extreme case where we have $k = 1$ cluster, indicating an infinitely coarse-grained latent learning, it trivially achieves perfect agree rates.

## D.2 Selecting Number of Clusters $k$

We conducted finer-grained experiments to examine the effect of the number of clusters $k$ on model performance, as shown in Figure 3, which offer empirical insights for selecting $k$. Based on the results, (1) a binary search approach could be used to efficiently find a suitable $k$, and (2) while it may require more computational resources, opting for a larger $k$ is generally a safe choice.

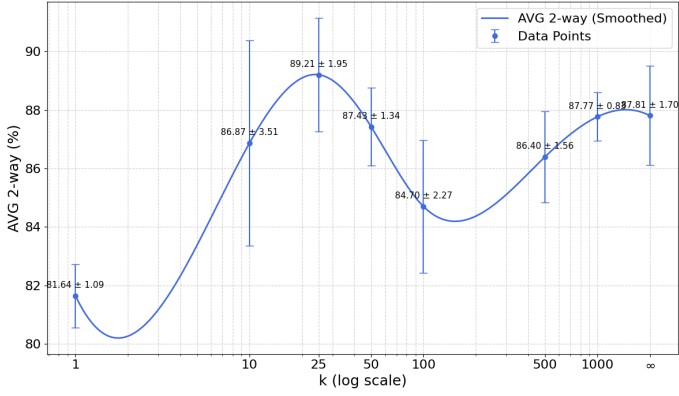

Figure 3: Smoothed model performance on Berka dataset regarding different $k$ (measured by AVG 2-way), where $k = \infty$ represents assigning each row a unique class.

## D.3 High-order Single-table Evaluation

We also consider higher-order single-table evaluation metrics for the quality of some representative tables as prior work [50]: 1) $\alpha$-precision and $\beta$-recall [1] to measure fidelity and diversity of synthetic data; 2) Machine Learning Efficacy (MLE) to measure the downstream-task utility; 3) Classifier Two Sample Test (C2ST) to measure if the synthetic data is distinguishable from real data by machine learning models.

| | PrivLava | SDV | ST-CTGAN | ST-TabDDPM | ST-ClavaDDPM | D-CTGAN | D-TabDDPM | D-ClavaDDPM | ClavaDDPM |
|---|---|---|---|---|---|---|---|---|---|
| **Household** | | | | | | | | | |
| $\alpha$-PRECISION | 97.79 ±1.18 | 87.32 ±0.06 | 83.23 ±1.71 | 85.68 ±0.09 | 99.83 ±0.02 | 91.59 ±0.02 | 90.92 ±1.38 | 90.94 ±0.17 | 99.77 ±0.00 |
| $\beta$-RECALL | 61.64 ±3.18 | 19.14 ±0.05 | 43.32 ±0.14 | 48.57 ±0.03 | 58.92 ±2.36 | 43.51 ±0.33 | 43.68 ±4.42 | 53.74 ±2.82 | 59.08 ±2.07 |
| C2ST | 97.06 ±1.68 | 85.56 ±0.00 | 77.54 ±0.70 | 66.68 ±0.16 | 99.60 ±0.00 | 68.04 ±0.02 | 63.93 ±5.45 | 71.19 ±0.02 | 99.55 ±0.13 |
| AVG F1 | 47.02 ±0.06 | 31.90 ±0.32 | 51.06 ±0.33 | 48.58 ±0.22 | 51.59 ±0.31 | 50.86 ±0.71 | 45.21 ±1.32 | 49.46 ±0.09 | 51.46 ±0.21 |
| AVG R2 | 67.89 ±0.02 | −18.24 ±0.19 | 63.00 ±0.70 | 65.96 ±0.22 | 66.31 ±0.74 | 64.07 ±0.19 | 64.24 ±1.50 | 66.33 ±0.00 | 66.71 ±0.59 |
| **Individual** | | | | | | | | | |
| $\alpha$-PRECISION | 99.44 ±0.02 | 55.88 ±0.08 | 83.56 ±0.07 | 88.41 ±0.01 | 99.74 ±0.06 | 87.52 ±0.01 | 96.98 ±0.24 | 99.55 ±0.15 | 98.69 ±0.50 |
| $\beta$-RECALL | 60.49 ±5.19 | 0.52 ±0.01 | 46.65 ±0.48 | 51.09 ±0.13 | 63.14 ±1.74 | 39.33 ±0.21 | 57.33 ±2.22 | 62.44 ±2.09 | 65.12 ±4.71 |
| C2ST | 99.63 ±0.25 | 5.29 ±0.00 | 78.32 ±0.73 | 68.23 ±0.08 | 99.76 ±0.17 | 76.44 ±0.12 | 94.50 ±1.46 | 99.23 ±0.01 | 97.36 ±0.19 |
| AVG F1 | 59.50 ±0.04 | 22.87 ±0.03 | 58.57 ±0.16 | 59.19 ±0.07 | 61.35 ±0.18 | 57.58 ±0.22 | 57.94 ±1.17 | 60.93 ±0.00 | 61.22 ±0.06 |
| AVG R2 | 73.51 ±0.30 | −167.24 ±6.42 | 80.52 ±0.30 | 81.53 ±0.18 | 83.13 ±0.00 | 77.91 ±0.97 | 82.23 ±0.43 | 83.08 ±0.03 | 83.12 ±0.07 |

Table 6: High-level single-table metrics evaluated on the *Household* table and the *Individual* table from the *California* dataset.

| | Household | Individual | Transaction | Order |
|---|---|---|---|---|
| DCR-SMOTE | 0.0295 | 0.0304 | 0.0082 | 0.0029 |
| DCR-ClavaDDPM | 0.0647 | 0.0407 | 0.0097 | 0.0101 |

Table 7: Median DCR comparison between ClavaDDPM and SMOTE.

We evaluated high-order single-table metrics on the *California* dataset across all baseline models and ClavaDDPM. Following [50], for the evaluation of MLE we perform a $90\%$, $10\%$ train-test split, where the $F_1$ and $R^2$ metrics are evaluated on the $10\%$ holdout set. Note that although PrivLava has an advantage on the *California* dataset when evaluated with statistical tests (Table 1), ClavaDDPM exhibits competitive, or even stronger performance than PrivLava on higher-order metrics. Especially for MLE, ClavaDDPM surpasses PrivLava by $13.07\%$ in terms of average $R^2$ in *Individual* table, and also beats PrivLava on average $F_1$ in both tables. Also notice that the baseline ST-CLAVADDPM dominates in high-order metric evaluations, demonstrating the strength of our Gaussian diffusion-only backbone model.

ClavaDDPM achieves a second-highest $\beta$-recall on *Household* table and ranks first in $\beta$-recall on *Individual* table with large margin, gaining a $7.65\%$ advantage over the best baseline without ClavaDDPM backbone. This serves as strong evidence that ClavaDDPM is not only data fidelity preserving, but is also capable of generating highly diverse data.

### D.4 Privacy Sanity Check

We follow TabDDPM [25] to perform a privacy sanity check against SMOTE [7], which is an interpolation-based method that generates new data through convex combination of a real data point with its nearest neighbors. We use the median Distance to Closest Record (DCR) [52] to quantify the privacy level. We compare the median DCR, as well as DCR distributions of ClavaDDPM against SMOTE on selected tables.

As shown in table 7, ClavaDDPM although neither specialized in privacy preserving, nor in single table synthesis, it still maintains a reasonable privacy level. The charts 4 demonstrates the distributions of DCR scores, where ClavaDDPM is in blue. The overall distribution is more leaning to the right side, indicating an overall higher DCR distribution.

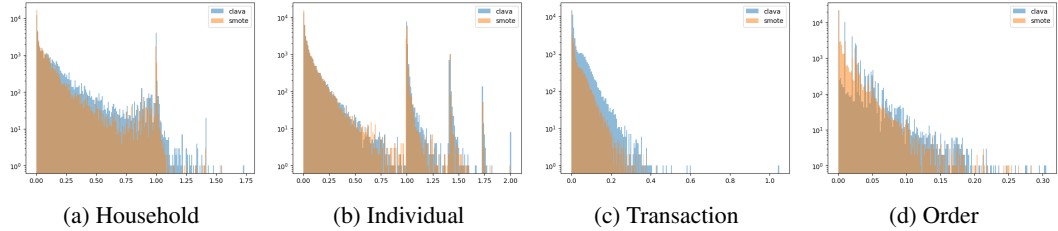

|        (a) Household        |        (b) Individual        |        (c) Transaction        |        (d) Order        |

Figure 4: DCR distributions of four selected tables. The $y$ axis is log-scaled for better presentation.

# E Complexity Analysis

Given a multi-relational database $\mathcal{G} = (\mathcal{R}, \mathcal{E})$, with $m$ tables, $n$ foreign key constraints, and $p$ rows per table. For a $p$-row table, we denote the time complexity of performing GMM clustering as $c_{\text{GMM}}(p)$, training a diffusion model as $c_{\text{diff}}(p)$, training a classifier as $c_{\text{class}}(p)$, synthesizing as $c_{\text{syn}}(p)$, ANN searching as $c_{\text{ANN}}(p)$.

**Phase 1: latent learning and table augmentation**

$$n \cdot c_{\text{GMM}}(p) \tag{13}$$

**Phase 2: training**

$$n \cdot c_{\text{class}}(p) + m \cdot c_{\text{diff}}(p) \tag{14}$$

Note that in practice this phase is dominated by diffusion training, primarily influenced by $m$.

**Phase 3: synthesis**

$$n \cdot c_{\text{syn}}(p) \tag{15}$$

**Additional step: matching**

$$n \cdot c_{\text{ANN}}(p) \tag{16}$$

Note that the runtime in this phase is negligible compared to the earlier phases, particularly with the FAISS implementation in the non-unique matching setup.

**Total**

$$n \left( c_{\text{GMM}(p) + c_{\text{class}}}(p) + c_{\text{syn}}(p) + c_{\text{ANN}}(p) \right) + m c_{\text{diff}}(p) \tag{17}$$

the overall runtime is dominated by Phase 2 (training) and Phase 3 (synthesis), with the critical factors being $m$, $n$, and $p$. The model remains robust against the number of clusters in Phase 1, as the impact on runtime is minimal due to the dominance of the later phases.

