# OpenReview forum: "ClavaDDPM: Multi-relational Data Synthesis with Cluster-guided Diffusion Models"
_NeurIPS.cc/2024/Conference — NeurIPS 2024 poster_

### Official Review · Reviewer_sDU8 · 2024-07-01

**Soundness:** 3
**Presentation:** 4
**Contribution:** 3
**Rating:** 7
**Confidence:** 3

**Summary:**

The paper tackles the problem of synthetic data generation in tabular formats and particularly focuses on data generation in multi-relational (multi-table) setups like relational databases. Whereas existing tabular diffusion models work only on single tables, the proposed ClavaDDPM extends the DDPM framework to support many relational tables linked via primary-foreign keys. In the hierarchy of parent-child tables, child tables contain keys from parent tables, and ClavDDPM conditions the generative process of child entries on their parent entries (grouped by the foreign key). To enable this, the model learns latent variables (expressed as GMMs) corresponding to the groups of foreign keys. Capturing the hierarchy of parent-child relations enables learning quite deep, long-range dependencies between k-hop neighbors in the relational schema. Experimentally, ClavaDDPM outperforms existing non-diffusion baselines both in terms of effectiveness and efficiency.

**Strengths:**

**S1.** The problem of synthetic data generation is of growing importance in the era of large, data-hungry models whose quality depends on the quality of training data. Besides, synthetic generation enables upsampling of a small custom dataset (with possible private and sensitive data) for fine-tuning backbone models. In the world of relational databases, most databases consist of multiple linked tables that have to be modeled jointly to capture data dependencies. ClavaDDPM is the first tabular DDPM that supports generation conditioned on multiple tables (organized in several hierarchy levels) and can be used on real-world datasets.

**S2.** The idea of learning latent variables with classifier guidance for grouped keys is rather clever. In addition, the model supports both numerical and categorical data as well as attempts to model the generative process for tables with several parents. Conditioning on several parent tables, the authors propose soft matching of latents (conditioned on each parent separately) via approximate nearest neighbor search.

**S3.** Compelling experimental results - ClavaDDPM was compared against existing non-diffusion multi-table models (PrivLava and Synthetic Dava Vault), the authors also adapted single-table baselines CTGAN and TabDDPM in several versions as baselines. Ablation studies are informative.

**S4.** The paper is well-written and structured, I enjoyed reading the manuscript.

**Weaknesses:**

**W1.** The idea of long-range dependencies in relational tables has been introduced in the beginning and highlighted in the experiments, but the main part (Section 4) does not go into the details. Perhaps Section 4.2 could expand on long-range modeling mapping to the example hierarchy from Figure 1.

**W2.** No discussion of the limitations, e.g., how long the training took; inference on the datasets takes several days. The checklist mentions “the second paragraph of the Conclusion” but this section in the manuscript has only one paragraph.

**Questions:**

**Q1.** Most of the reported numbers (total variation distance between synthetic and real data) are in the higher 95%+ range which makes it hard to distinguish the quality of new models. Are there any other metrics close to real-world tasks that could be of use? For instance, TabDDPM and others report the quality of CatBoost/XGBoost and other standard tabular ML algorithms trained on the synthetic data.

Comments:
* Line 206: Typo in LavaDDPM
* Captions of Tables 1 and 2 could also mention what the reported metrics are.

**Limitations:**

There is no discussion of the limitations.

---

> ### Author Rebuttal · Authors · 2024-08-06
>
> **A-W1 Metric Details:**
>
> Thank you for your suggestion. We agree that expanding on the details about long-range dependencies in Section 4.2 and mapping them to the example hierarchy from Figure 1 would greatly enhance the clarity. We will make this change in the revised version.
>
> $\quad$
>
> **A-W2 Limitations and Future Works:**
>
> Thank you for identifying this issue, and we apologize for the confusion.To respect NeurIPS page limit, we submitted a shortened version of our paper. However, after the submission deadline, we realized that the limitations and future work that we referred to were unfortunately part of what was cut . Here is the original second paragraph of the conclusion, we will make sure to add this to the main paper in the revised version:
>
> >"We focused on foreign key constraints in this work, and made the assumption that child rows are conditionally independent given corresponding parent rows. This brings three natural follow-up research directions: i) extension to the scenarios where this prior information is not available and these relationships need to be discovered first[3], ii) further relaxing the assumptions, and iii) inspecting multi-relational data synthesis with other integrity constraints (e.g, denial constraints[4], general assertions for business rules).
>
> >Furthermore, we evaluated ClavaDDPM's privacy with the common (in tabular data literature) DCR metric. Nonetheless, we think it is worthwhile to: i) evaluate the resiliency of ClavaDDPM against stronger privacy attacks[5], and ii) investigate the efficacy of boosting ClavaDDPM with privacy guarantees such as differential privacy. Similarly,  the impacts of our design on fairness and bias removal, as another motivating pillar in synthetic data generation, is well worth exploring as future work. We believe the thorough multi-relational modeling formulation we presented in this work, can serve as a strong foundation to build private and fair solutions upon.
> "
>
> $\quad$
>
> **A-Q1 Real-world Metrics:**
>
> Yes, there are other metrics being used for evaluation. We conducted single-table machine learning efficacy (MLE) experiments using the same settings as in TabSyn, which employs an XGBoost model to predict one column given the remaining columns. The results of these experiments are presented in the appendix (section D.2). These results demonstrate that, in terms of high-order or real-world metrics, ClavaDDPM achieves state-of-the-art performance, even though it was designed for multi-table synthesis. In the revised version of our paper, we will include comments about the MLE results in the main paper.
>
> Additionally, our primary focus in this work is on multi-table quality. As shown in Table 1, ClavaDDPM exhibits significant advantages over other baselines regarding long-range dependencies. This advantage becomes more pronounced as the number of hops increases, making the quality of ClavaDDPM more distinguishable.
>
> For multi-table MLE, this remains an unexplored area, and we plan to address it in our future work.
>
> $\quad$
>
> **Reply to comments:**
>
> Thank you for pointing out the typo, and suggestions about captions. We will make these changes in the revised version.

---

> > ### Comment · Reviewer_sDU8 · 2024-08-09
> >
> > Thank you, after reading the responses to this and other reviews my concerns are resolved and I remain positive about the work.

---

> > > ### Author Response · Authors · 2024-08-11
> > >
> > > Thank you for your time and effort in reviewing our work as well as the rebuttal. We greatly appreciate your constructive feedback and the positive score you've given, which encourages us to further refine the research.

---

### Official Review · Reviewer_bZ6q · 2024-07-11

**Soundness:** 2
**Presentation:** 3
**Contribution:** 2
**Rating:** 4
**Confidence:** 3

**Summary:**

This paper proposes the new ClavaDDPM approach to address the scalability and long-range dependency challenges in tabular data synthesis. ClavaDDPM uses clustering labels to model inter-table relationships, particularly focusing on foreign key constraints, and employs diffusion models' robust generation capabilities along with efficient algorithms to propagate learned latent variables across tables. Evaluations demonstrate ClavaDDPM's superior performance in capturing long-range dependencies and competitive utility metrics for single-table data.

**Strengths:**

$\bullet$ The manuscript is commendably well-structured, with a logical flow that effectively guides the reader through the paper. The clarity in writing and the organization of content significantly enhance the reader's comprehension and the overall quality of the presentation.

$\bullet$ The authors have conducted a thorough experimental validation, which is a significant strength of the paper. The meticulous detailing of experimental procedures, including data handling, model training, and evaluation metrics, greatly contributes to the reproducibility and credibility of the study.

$\bullet$ The topic of the paper addresses a highly relevant issue in the field, which is a strong point. The research aligns well with current trends and challenges in the domain, ensuring that the findings will be of interest to a broad audience and have potential applications in real-world scenarios.

**Weaknesses:**

$\bullet$ The manuscript exhibits a lack of punctuality and standardization in the presentation of mathematical formulas, leading to potential ambiguity in interpretation. Key terms are not consistently defined upon first use, and there is a misuse of symbols that may confuse readers. Additionally, the placement of table names appears to be incorrect in some instances, further detracting from the paper's clarity. It is essential to standardize the notation and ensure consistent definitions and proper placement of table names for better readability.

$\bullet$ The assumptions made in the model section require more rigorous justification. The authors should provide a more detailed explanation of why these assumptions are reasonable and how they contribute to the validity of the model. This will strengthen the theoretical foundation of the work and enhance the credibility of the proposed approach.

$\bullet$ The paper would benefit from an analysis of the model's complexity, particularly in the context of large-scale databases. Given the increasing size and complexity of real-world databases, understanding how the proposed model scales and performs under such conditions is crucial. Including a complexity analysis will provide insights into the model's practicality and efficiency.

$\bullet$ The ablation study regarding the cluster number 'k' lacks necessary granularity. A more fine-grained investigation is needed to understand the impact of varying 'k' on the model's performance and to identify the optimal settings for different scenarios. This will provide a clearer understanding of the model's behavior and its adaptability to various contexts.

$\bullet$ The section on the multi-relational synthesis problem, currently placed in the appendix, should be relocated to the main body of the paper. This will improve the flow of the paper and make it easier for readers to follow the synthesis process and understand its relevance to the overall work. Integrating this section into the main text will enhance the coherence and comprehensiveness of the paper.

**Questions:**

See the above weaknesses.

**Limitations:**

No.

---

> ### Author Rebuttal · Authors · 2024-08-06
>
> **A1: Presentation improvements**
>
> Thank you for your valuable feedback. We will take steps to improve the writing and overall presentation of the paper, and have fixed the table names issue, and will have a better written revised version.
>
> To further enhance our manuscript, we would greatly appreciate it if you could point out any specific issues with the mathematical formulations or inconsistency in definitions, so we can address them in detail.
>
> $\quad$
>
> **A2: Justification of assumptions**
>
> Thank you for highlighting the need for more rigorous justification of our assumptions. We appreciate your feedback and would like to provide a detailed explanation:
> We understand the importance of validating assumptions in tabular data synthesis. When reviewing the literature, such as TabSyn and TabDDPM, we observed that they implicitly rely on the naive row-wise i.i.d. assumptions, which lack validity when extended to multi-table scenarios. To address this, we reexamined the problem and adopted a weaker and more realistic assumption: that child rows are conditioned on foreign key constraints. We provided a mathematical deduction to strengthen the theoretical foundation of our approach, aiming to address tabular data synthesis more rigorously than simply applying generative models as done in previous works.
> While it is challenging to justify these assumptions perfectly in alignment with real-world scenarios, we designed experiments to empirically validate them. In section 5.1, Table 1, we compared ST-ClavaDDPM with ClavaDDPM. ST-ClavaDDPM utilizes the same model backbone as ClavaDDPM but makes the i.i.d. assumption on all child table rows. In contrast, ClavaDDPM assumes that child rows are conditioned on parent rows. The experimental results in Table 1 indicate that ClavaDDPM significantly outperforms ST-ClavaDDPM, empirically demonstrating that our assumption is more effective on real-world datasets compared to the naive row-wise i.i.d. assumption.
> Although our assumptions may not be perfect, our empirical results show that the theoretical analysis in our paper is robust and surpasses the assumptions made in previous works. Additionally, we consider exploring even weaker and more realistic assumptions as a direction for future work.
>
> $\quad$
>
> **A3: Complexity analysis**
>
> Thank you for your suggestion. We agree that a complexity analysis is essential for understanding the scalability and efficiency of our model, especially for large-scale databases. Here is a preliminary analysis, which we will expand upon with detailed results in the revised version.
>
> Given a multi-relational database $G = (R, E)$ with $m$ tables, $n$ foreign key constraints, and $p$ rows per table. Given a p-row table, we denote the time complexity of performing GMM clustering $c_{GMM}(p)$, training a diffusion model as $c_{diff}(p)$, training a classifier as $c_{class}(p)$, synthesizing as $c_{syn}(p)$, ANN searching complexity as $c_{ann}(p)$.
>
> *Phase 1*: Latent Learning and Table Augmentation:
> - Runtime: $n \cdot c_{GMM}(p)$.
>
> *Phase 2*: Training:
> - Runtime: $n \cdot c_{class}(p) + m \cdot c_{diff}(p)$.
> - This phase is dominated by diffusion training, primarily influenced by $m$.
>
> *Phase 3*: Synthesis:
> - Runtime: $n \cdot c_{syn}(p)$.
>
> *Additional Step*: Matching:
> - Runtime: $n \cdot c_{ann}(p)$.
> - Negligible runtime with FAISS implementation.
>
> *Summary*:
> - Dominated by Phase 2 (training) and Phase 3 (synthesis).
> - Critical factors: $m, n$, and $p$.
> - Robust against the number of clusters $k$ in Phase 1 due to the dominance of later phases.
> Our model shows significant scalability and practicality compared to existing methods like SDV, which is limited to synthesizing at most 5 tables with a depth of 2 (section 5.2). We will provide empirical runtime measurements and baseline comparisons in the revised version.
>
> $\quad$
>
> **A4: Fine-grained investigation of $k$**
>
> Thank you for your valuable suggestion. We agree that a more fine-grained analysis of the cluster number $k$ is essential for understanding its impact on the model's performance and identifying optimal settings for different scenarios.
> Our preliminary results show an interesting trend: the model quality initially performs lower when $k$ is small, peaks at $k=25$, and then decreases. When $k$ is very large, the model performance converges to a local optimum. This suggests that a technique similar to binary search could be applied empirically to find the optimal $k$.
>
> $\quad$
>
> $\textbf{Table 1: Model Performance for Different k}$
> |Berka|$k=1$|$k=10$|$k=25$|$k=50$|$k=100$|$k=500$|$k=1000$|$k=\infty$|
> |-|-|-|-|-|-|-|-|-|
> |AVG 2-way|81.64 $\pm$ 1.09|86.87 $\pm$ 3.51|89.21 $\pm$ 1.95|87.43 $\pm$ 1.34|84.7 $\pm$ 2.27|86.4 $\pm$ 1.56|87.77 $\pm$ 0.83|87.81 $\pm$ 1.70|
>
> The results of these experiments will be included in the revised paper. This will provide a clearer understanding of the model's behavior and its adaptability to various contexts.
>
> $\quad$
>
> **A5: Relocation of research problem**
>
> Thank you for pointing this out. We agree that moving the section on the multi-relational synthesis problem to the main body of the paper would improve the flow and make it easier for readers to follow the synthesis process and understand its relevance to the overall work.
> However, due to the NeurIPS page limit, we had to place the definition in the appendix. We will integrate this section into the main text in the revised version to enhance the coherence and comprehensiveness of the paper.

---

> > ### Comment · Area_Chair_5E22 · 2024-08-12
> >
> > Dear reviewer,
> >
> > Please respond to the rebuttal and provide substantive arguments and, in particular, more concrete strengths and weaknesses of the paper. Unless you update your line of argument, I will not be able to take your review into account in the decision making process.

---

> > ### Comment · Reviewer_bZ6q · 2024-08-13
> >
> > Thank you for your detailed responses and efforts to address my concerns. I appreciate the clarifications provided, particularly regarding the assumptions in your approach. However, after careful consideration, I have decided to maintain my original score. I believe there are still several areas that require further attention:
> >
> > Reference to Appendix A Notation Summary: I suggest adding explicit references to the notation summary provided in Appendix A within the main text. This will help readers navigate the mathematical formulations more effectively and avoid potential confusion.
> >
> > Issues with Equations 5-9: I noticed that there are some missing symbols in Equations 5 through 9, which result in these equations not being properly formatted. Addressing these inconsistencies is crucial for the clarity and accuracy of your mathematical presentation.
> >
> > Visualization of Label Clustering and Table Relationships: It would be beneficial to include some visualizations that illustrate the relationship between label clustering and table structures. Such visual results would provide additional empirical evidence and enhance the understanding of your approach.
> >
> > Clarification on Mathematical Deduction: You mentioned that you provided a mathematical deduction to strengthen the theoretical foundation of your approach. Could you clarify where and how this deduction is presented in the paper? A clearer explanation or more detailed description would be helpful in understanding the rigor of your theoretical contributions.
> >
> > I hope these suggestions help further refine your work. Thank you again for your dedication and for considering my feedback. I look forward to seeing the final version of your paper.

---

> > > ### Author Response · Authors · 2024-08-13
> > >
> > > Thank you for your response and we appreciate your comments that help refine our paper.
> > >
> > > $\quad$
> > >
> > > > Reference to Appendix A Notation Summary
> > >
> > > Thank you for your feedback. To ensure clarity and consistency, we have included a paragraph at the beginning of Section 4.1 that introduces the relevant notations, such as parent/child tables, the entire database, and the distinctions between data and random variables. This section is complemented by a running example using a subset of the Berka dataset. We appreciate your suggestion and will add a reference to Appendix A for further clarification.
> > >
> > > $\quad$
> > >
> > > > Issues with Equations 5-9
> > >
> > > We would greatly appreciate it if you could point out any specific missing symbols or inconsistencies in Equations 5-9. This would help us further improve the clarity and accuracy of our work.
> > >
> > > $\quad$
> > >
> > > >Visualization of Label Clustering and Table Relationships
> > >
> > > Thank you for your suggestions. We will consider adding some visualizations in the appendix to enhance understanding in the revised version.
> > >
> > > $\quad$
> > >
> > > >Clarification on Mathematical Deduction
> > >
> > > The mathematical derivation spans from Equation 5 to Equation 10, where we derive a probability expression for the parent-child scenario under the assumptions we have applied. The final expression (Equation 10) illustrates the distribution modeled by each component in the ClavaDDPM framework. We believe this strengthens the theoretical foundations of our work, as previous approaches in tabular data synthesis, such as TabDDPM and TabSyn, do not explicitly provide the probability assumptions they rely on or clarify the distributions they model, instead solving the problem in a purely end-to-end manner.
> > >
> > > - Equation 5 is a direct result of the i.i.d. assumption on $(g_j, y_j)$.
> > > - Equation 6 is based on conditional probability.
> > > - Equation 7 introduces an independence assumption we are making.
> > > - Equation 8 is a series of conditional probability expansions of Equation 6, given Equation 7.
> > > - Equation 9 expands the conditional foreign key group distribution by introducing the group size variable, which directly extends a part of Equation 8.
> > > - Equation 10 presents the final expression of the parent-child distribution we are modeling, incorporating the independence assumptions introduced in our work.
> > >
> > > This final expression involves three probability distributions, each representing a learnable component of the ClavaDDPM framework:
> > > - $p(y, c)$ is the augmented parent distribution, modeled by training a parent diffusion model.
> > > - $p(s|c)$ is the conditional group size distribution, calculated through frequency counting.
> > > - $p(x|c)$ is the conditional child distribution, modeled using classifier-guided diffusion.
> > >
> > > We are happy to provide further details and welcome any specific recommendations to improve clarity.

---

### Official Review · Reviewer_5kZH · 2024-07-12

**Soundness:** 2
**Presentation:** 2
**Contribution:** 2
**Rating:** 3
**Confidence:** 3

**Summary:**

This paper proposes ClavaDDPM to address two key deficiencies in multi-table data generation: scalability for larger datasets and capturing long-range dependencies, such as correlations between attributes across different tables. This approach utilizes cluster labels as intermediaries to model relationships between tables, paying special attention to foreign key constraints. The authors elaborated on the proposed method, made some assumptions, conducted some experiments for evaluation, and performed some analysis about the results.

**Strengths:**

1. The paper proposes an efficient framework to generate multi-relational data that preserves long-range dependencies between tables, and proposes relation-aware clustering for modeling parent-child constraints.

2. The paper applies a matching technique based on approximate nearest neighbor search as a general solution to the problem of multi-parent relation synthesis for a child table with multiple parents.

**Weaknesses:**

1. I have some concerns about the assumptions made in the paper, which seem to be somewhat inconsistent with the scenarios in real world settings. Please refer to Questions 1 and 2 for details.

2. There may be some clarifications needed in the experimental section. Meanwhile, the analysis of the experimental results does not cover some key points. Please refer to Questions 3-6 for details.

3. There are some problems with the writing and formatting of this paper. Please refer to Questions 7-8 for details. Also, there are some confusing statements in the paper, which hinder the readability of the paper. Please refer to Questions 9-10 for details.

**Questions:**

1. The paper assumes that there are parent-child relationships between different tables and the relationships are known. I However, in real life, the relationships may be very complex and may not be known in advance. In this case, how can ClavaDDPM handle it?

2. The i.i.d. assumption in Section 4.1 seems too strong, because the child rows corresponding to different primary keys may not be independent in real-world scenarios. If this is the case, does the theoretical analysis in the paper still hold?

3. Please clarify why you selected CTGAN over newer or state-of-the-art models like CTAB-GAN, TabSyn, and CoDi mentioned in the related work for your experiments.

4. The reason for the absence of average agree rate in the last two columns of Table 2 remains unclear. Additionally, it is not specified what NA represents. Moreover, there is a lack of clarity regarding the calculation method and formula used to determine the agree rate in this experiment.

5. I feel a bit confused about why TabDDPM converges so slowly on the Instacart05 and Movie Lens datasets. The authors of TabDDPM stated in their work that TabDDPM can converge quickly with a 2080Ti GPU. In addition, I noticed that the Berka dataset seems to be larger than those two datasets. Could you kindly provide an explanation as to why TabDDPM exhibits convergence on this larger dataset, while failing to do so on the two smaller datasets?

6. Are the evaluation metrics used in the experiment widely used in the field or are they proposed by the authors? I haven't seen similar metrics in related works. If they are proposed by the authors, why don't you use metrics from other papers, such as the relative error used in PrivLava [1] or more comprehensive metrics used in TabSyn [2]?

7. There are some typos in the paper: “LavaDDPM” in line 206, “Dnorm” in the caption of Table 1, “The experiment result show…” in line 355.

8. Page 18's experimental results table lacks a caption.

9. In Section 5.2, you referred to additional results in Appendix 5.3, which is missing from the paper. Please provide the mentioned experimental results.

10. In the checklist, you said that you discussed the limitations and future work in the second paragraph of the conclusion section, but there is no relevant discussion in the conclusion of the paper, not even the second paragraph, which is confusing.

Reference:

[1]	K. Cai, X. Xiao, and G. Cormode. Privlava: synthesizing relational data with foreign keys under differential privacy. Proceedings of the ACM on Management of Data, 1(2):1–25, 2023.

[2]	H. Zhang, J. Zhang, B. Srinivasan, Z. Shen, X. Qin, C. Faloutsos, H. Rangwala, and G. Karypis. Mixed-type tabular data synthesis with score-based diffusion in latent space. arXiv preprint arXiv:2310.09656, 2023.

---

> ### Author Rebuttal · Authors · 2024-08-06
>
> **A1: Known relationships**
>
> We fully agree on the importance of realistic assumptions. This work tackles a prevalent challenge in the finance industry [in collaboration with them] on tabular synthesis. Synthesizing data for multiple interconnected tables, even with known foreign keys, has been entirely overlooked in the literature, with only two notable exceptions offering inefficient solutions [1,2]. Our work presents a practical approach that surpasses the state of the art (SOTA) with known foreign keys. We will discuss addressing unknown constraints in future work in the revision.
>
> $\quad$
>
> **A2: I.i.d assumptions**
>
> We appreciate your perspective and agree that the assumption in Sec 4.1 is strong. While previous works TabSyn and TabDDPM [6,7] made even stronger assumptions (each row is i.i.d), we introduced a weaker assumption: in a parent-child two-table scenario (in Sec 4.1), each parent row is i.i.d., while the child rows are dependent on its corresponding parent row. In fact, we are making a Bayesian modeling assumption that each child row, although not independent by itself, is conditionally independent of other child rows given its parent.
>
> Our evaluation in Sec 5.1 demonstrates the effectiveness of our weaker assumption (ClavaDDPM) on real-world datasets compared to that of prior works (ST-ClavaDDM with strong i.i.d assumptions for child rows).
> While our assumption may not be perfect (modeling all dependencies with perfect accuracy is hard), our empirical results support our theoretical analysis and outperform previous works' assumptions. We will clarify these and discuss weaker and more realistic assumptions as future work.
>
> $\quad$
>
> **A3: Choice of baselines (CTGAN and TabDDPM over CTABGAN, TabSyn, and CoDi)**
>
> Our work focuses on a multi-table synthesis with a new paradigm. Thus, the strength of single-table baseline models was not our only consideration. Our goal was to pick one diffusion model and one GAN-based model, each being *SOTA* and *representative* in their domain for a diverse comparison.
> Firstly, we excluded CTABGAN(+) or CoDi because they were shown inferior to TabDDPM and TabSyn [6,7]. Since both TabDDPM and TabSyn are SOTA, we chose between them. Given our ClavaDDPM employs a similar model to TabDDPM, comparing against baselines using TabDDPM offers a fair evaluation, highlighting the effect of our Clava framework while disregarding the inherent advantages of TabSyn over TabDDPM. Additionally, our preliminary results (with slightly different metrics) at Clava’s early development indicated TabDDPM’s superior performance over TabSyn as the backbone on real-world datasets. **(Table 2 in comments)**
>
> Thus, we chose TabDDPM as the backbone for both our model and the two baselines, ST- and Dnorm-. Even with other baselines, ClavaDDPM's advantage would remain evident.
>
> Though CTGAN is weaker than TabDDPM,  we included it because it is a representative GAN-based tabular synthesis model, aligning with the TabSyn paper’s inclusion of CTGAN over CTABGAN.
> We will add the clarification and include experiments on other baseline models, as you suggested in the revised version.
>
> $\quad$
>
> **A4: Clarification on agree rate**
>
> Thanks for the suggestion. We will detail the agree rate formulation in the appendix. The agree rate is only influenced by k and lambda, but not eta or “no matching” (the last two columns that are not involved in the clustering process). We will include the agree rates in those columns for completeness and clarity.
>
> $\quad$
>
> **A5: Clarification on convergence**
>
> We will add a complexity analysis in the revised paper. Convergence depends on both the data size and the domain size. Even though TabDDPM can handle all datasets on a 2080Ti GPU (A6000 is overkill), its multinomial diffusion for categorical features is a bottleneck with large domain sizes, as exemplified by  Instacart05 and Movie Lens datasets. For instance, Instacart05 has a categorical column with 49,688 values, requiring one-hot encoding and expensive multinomial diffusion. In TabDDPM's single-table datasets, categorical columns have relatively few values(e.g., the largest category count in the adult dataset is 42). The Berka dataset, despite having more tables, has the largest categorical domain size of 77, allowing faster convergence for TabDDPM.
>
> Real-world data often contain categorical columns with numerous distinct values. ClavaDDPM accelerates training by applying a unified Gaussian diffusion to both categorical and numerical columns, leading to faster convergence.
>
> Additionally, for fair comparison, we aligned TabDDPM's hyperparameters with ClavaDDPM's, which are larger than the original TabDDPM settings, and thus will be slower than the original TabDDPM (e.g., diffusion timesteps = 2000 vs. 1000).
>
> Our TabDDPM experiments used the TabDDPM library within the Synthcity framework. We will release the baseline code for reproducibility.
>
> $\quad$
>
> **A6: The choice of evaluation metrics**
>
> We used single-table data quality metrics from TabSyn and reported KS/TV metrics in the main paper, and the rest (alpha precision, beta recall, classifier detection, machine learning efficacy) in the appendix (section D.2) due to space constraints.
>
> We did not use the multi-table metric like relative error from PrivLava for
> - It relies on human-designed queries, not applicable to all datasets.
> - No open-source evaluation code, hard to replicate.
> - No theoretical validation for these hand-designed queries.
>
> These limitations highlighted a gap in multi-table synthesis evaluation. Therefore, we proposed the long-range dependency metric,  a statistical measure applicable to all multi-table datasets without needing hand-designed queries.
>
> $\quad$
>
> **A 7,8,9: Formatting issues**
>
> Thank you for your feedback. We will address them in our revision.
>
> $\quad$
>
> **A10: Conclusion**
>
> We apologize for the omission of limitations and future work due to page constraints. These sections will be included in the revision.

---

> > ### Comment · Area_Chair_5E22 · 2024-08-12
> >
> > Dear reviewer,
> >
> > Please respond to the rebuttal as is expected of you as a NeurIPS reviewer asap! Thanks

---

> > ### Author Response · Authors · 2024-08-14
> > **Additional comment on question 9**
> >
> > We would like to additionally address question 9, where the mentioned experiment results are actually located in Appendix D.1. It was a typo for reference and we will fix it in the revision.

---

### Official Review · Reviewer_ok29 · 2024-08-08

**Soundness:** 3
**Presentation:** 2
**Contribution:** 3
**Rating:** 5
**Confidence:** 3

**Summary:**

This work addresses the challenges inherent in generating multi-relation tabular data and proposes a novel generation method based on hidden random variables.  The approach analysis correlations between primary and foreign keys between tables by predicting hidden variables associated with these keys within simultaneous tables.  These hidden random variables are inferred using Denoising Diffusion Probabilistic Models (DDPM) grounded in stochastic processes.  The method is elaborated from two-table linkage generation to multi-table linkage generation, with a step-by-step explanation of the approach.  Comprehensive experiments are conducted to demonstrate the practical effectiveness of the proposed method.

**Strengths:**

This paper highlights the extensive dependencies between different tables in the process of multi-relation tabular data synthesis and introduces an innovative approach by analyzing the relationships between primary keys and foreign keys using hidden variables.  The proposed ClavaDDPM process is well-articulated and has demonstrated promising results across several datasets.  Additionally, the subsequent analysis of hidden variables enhances the credibility of this work.

**Weaknesses:**

Although this work attempts to elaborate on its theoretical framework with substantial content, the notation system and descriptive symbols used do not conform to standard academic mathematics conventions (see sections 3. Background and 4.1 Notations). The marks in equation are not uniform (Eq.2). Furthermore, several key formulas lack thorough proofs (for example, Formula 8). The proof of this key formula would be best placed in the appendix. Additionally, Formula 3 from a previous study lacks supporting evidence; a brief explanation of its proof process or an indication of its rationale is necessary. The validity of the method “we introduce latent random variables c such that g is independent of y conditioned on c” also requires theoretical verification.

**Questions:**

Question 1: In line 79 to 85, the author mention table with R and relation with R. Is this means that a relation in Relational Database is a table in this paper? Since the relations are model as edges of graph.

Question 2: According to the paper content, the author mention that “the primary key of table serves as the unique identifier for each row in the table”. And in section 4.1 the rows in parent table can suitable i.i.d assumption. If every row is unique, how rows can be suitable independent and identification distributions?

Question 3: From the perspective of learning latent variable method alone, what are the advantages of CLavaDDPM proposed in this paper compared with other studies?

**Limitations:**

The authors adequately addressed the limitations.

---

> ### Author Response · Authors · 2024-08-09
> **Rebuttal to the Questions**
>
> **A1: Terminologies of table**
>
> Yes, a relation is a table. In this paper, we primarily adhere to the terminology used in relational database literature, where a “table” is formally referred to as a “relation”, and multi-relational synthesis corresponds to multi-table synthesis. Consequently, our notation slightly differs from previous works that only consider single tables, such as TabDDPM[2] and TabSyn[3]. To be clear, we used “tables” and “relations” interchangeably in the paper. Specifically, we model “relations” or “tables” as nodes and “foreign key constraints” as edges, consistent with relational database terminology. We apologize for any confusion and will clarify this further in the revised version.
>
> $\quad$
>
> **A2:I.i.d. assumptions**
>
> We fully understand your concern. In our work, the term “unique identifier” refers to a unique ID for a row  and is not related to the data distribution itself. Below is an example table:
>
> |Primary key(id)|Gender|Country|
> |-|-|-|
> |2|M|US|
> |3|F|UK|
>
> In this table, the primary key (unique identifier) is simply an ID, while the actual data distribution involves attributes like (gender, country). By i.i.d., we mean that the distribution of $(gender, country)$ is independent. When modeling, the primary key is not included as part of the data but is used solely for enforcing foreign key constraints. For example, if a child table row has a foreign key value of $3$, it refers to the row $(F, UK)$ in the parent table. The value $3$ is not passed into the diffusion model and is assigned additionally.
>
> Similarly, previous works in tabular data synthesis, such as SDV, PrivLava, and TabDDPM, treat keys as identifiers rather than data, with key values not being passed into the model. We will clarify this in our revised version to prevent further confusion.
>
> $\quad$
>
> **A3: Latent variable learning**
>
> In the context of tabular data synthesis, PrivLava[4] is the only prior work that has learned latent variables (excluding the works that directly adopt latent generative models, which do not explicitly learn latent variables). Compared to PrivLava, our approach is agnostic to data domain size and has faster convergence. PrivLava's latent learning is marginal-based, making it applicable only to predefined fixed domains (they primarily support integers). Additionally, their use of the EM algorithm is slow and prone to convergence issues in practice. Our experiments in Section 5, Table 1, demonstrate that their method fails to converge on many real-world datasets.
>
> Importantly, our method (diagonal covariance GMM) enforces our independence assumption and aligns with our theoretical analysis. In Section 4.1, Equation 7, we assume that child table groups $g$ are independent of parent row $y$, conditioned on latent variable $c$. Thus, we learn latent variable $c$ such that its distribution supports this assumption. As mentioned in Section 4.3.1, we use a GMM with diagonal covariances in the joint space of $(X, Y)$. This ensures that, when the GMM is properly trained, the covariance between $X$ and $Y$ is zero in each Gaussian cluster with centroid $c$. Consequently, conditioned on $c$, $g$ and $y$ will tend to be independent, as $g$ is a collection of $X$.
>
> While more advanced latent learning methods, such as VQVAE[5], could perform latent learning and quantization simultaneously, we considered them during development but identified potential trade-offs in terms of runtime. Currently, the runtime of the latent learning phase is negligible compared to other phases, and VQVAE lacks synergy with our theoretical assumptions. Nevertheless, as noted at the end of Section 4.3.1, we are open to exploring the impact of other latent learning methods in future work.

---

> ### Author Response · Authors · 2024-08-09
> **Rebuttal to the Weakness**
>
> >Although this work attempts to elaborate on its theoretical framework with substantial content, the notation system and descriptive symbols used do not conform to standard academic mathematics conventions (see sections 3. Background and 4.1 Notations).
>
> **R: Non-standard notations**
>
> Thank you for pointing this out. We acknowledge that our notation differs slightly from the standard. In probability theory, capital letters typically denote random variables, while lower case letters represent their actual values. We modified this convention to distinguish between row data and table data. For example, we use $x$ to denote a row of a child table and $X$ to denote the entire child table. To avoid conflicts, we use bold letters $\mathbf{x}$ and $\mathbf{X}$ to represent the corresponding random variables.
> To clarify this further, we will describe our design choices in detail and ensure consistency throughout the manuscript in the revised version.
>
> $\quad$
>
> >The marks in equation are not uniform (Eq.2)
>
> **R: Non-uniform marks**
>
> We obtained Equation 2 from the Background section of TabDDPM[2]. We would greatly appreciate it if you could point out the specific issues with the notation or formality, and we are more than happy to make any necessary improvements.
>
> $\quad$
>
> >Furthermore, several key formulas lack thorough proofs (for example, Formula 8). The proof of this key formula would be best placed in the appendix.
>
> **R: Proofs for formula**
>
> We agree with your observation that this is a key formula in our work. The derivation is based on Equation 7. Due to space constraints, we provided only a brief one-line expansion in the main paper. We appreciate your suggestion and will include a detailed proof in the appendix.
>
> $\quad$
>
> >Formula 3 from a previous study lacks supporting evidence; a brief explanation of its proof process or an indication of its rationale is necessary.
>
> **R: Formula 3**
>
> Formula 3 is directly adopted from previous work [1], which was derived using Bayes' theorem. Due to space limitations, we were unable to provide a more detailed explanation in the main paper and included only the result. However, we appreciate your point and will add a more detailed background section in the appendix of the revised version. Additionally, we will briefly mention the Bayesian modeling rationale in the main paper.
>
> $\quad$
>
> >The validity of the method “we introduce latent random variables c such that g is independent of y conditioned on c” also requires theoretical verification.
>
> **R: Validity of assumption**
>
> This is a valid concern. In fact, ClavaDDPM adopts a Bayesian modeling paradigm, where we make certain Bayesian assumptions and let the trained models enforce those assumptions. Therefore, we chose GMM with diagonal covariance, which by design learns latent variables $c$ that enforce conditional independence between $g$ and $y$ (with covariance being zero). We will include a more detailed demonstration of this aspect in the appendix to better address the theoretical foundations.

---

> ### Author Response · Authors · 2024-08-09
> **References**
>
> [1] P. Dhariwal and A. Nichol. Diffusion models beat gans on image synthesis. Advances in neural information processing systems, 34:8780–8794, 2021.
>
> [2] A. Kotelnikov, D. Baranchuk, I. Rubachev, and A. Babenko. Tabddpm: Modelling tabular data with diffusion models. In International Conference on Machine Learning, pages 17564–17579. PMLR, 2023.
>
> [3] H. Zhang, J. Zhang, B. Srinivasan, Z. Shen, X. Qin, C. Faloutsos, H. Rangwala, and G. Karypis. Mixed-type tabular data synthesis with score-based diffusion in latent space. arXiv preprint arXiv:2310.09656, 2023.
>
> [4] K. Cai, X. Xiao, and G. Cormode. Privlava: synthesizing relational data with foreign keys under differential privacy. Proceedings of the ACM on Management of Data, 1(2):1–25, 2023.
>
> [5] Razavi, Ali, Aaron Van den Oord, and Oriol Vinyals. "Generating diverse high-fidelity images with vq-vae-2." Advances in neural information processing systems 32 (2019).

---

### Author Rebuttal · Authors · 2024-08-06

We greatly appreciate the reviewers' efforts and constructive feedback. We humbly accept their suggestions and will make improvements accordingly. These insights will help make the paper more solid and better organized. Here, we address some general questions and misunderstandings and outline the steps we will take to address the issues in each aspect.

$\quad$

## Assumptions and Theoretical Foundations

>Previous works like TabDDPM and TabSyn on tabular data synthesis for single tables have implicitly assumed that each row in the table is i.i.d. (independently and identically distributed). This assumption: (1) has not been specifically analyzed or validated, and (2) is unsuitable for multi-table synthesis. Therefore, our work aims to address this by extending the problem to multi-table synthesis. Our improvements include:
>1. Explicitly stating the assumptions and deriving mathematical deductions based on them.
>2. Introducing a weaker, more realistic assumption for multi-table synthesis:
>    - In a multi-table database forming a DAG, each row in a root table (i.e., tables without parent tables) is i.i.d.
>    - Each child table row is dependent on its corresponding parent row.
>
>We acknowledge that misunderstandings occurred, and we will enhance clarity and organization in the revision.

$\quad$

## General Writing and Formatting

>Reviewers identified several typos and incorrect table references. Additionally, we received valuable suggestions to move some sections from the appendix to the main paper and to add discussions of limitations and future work. These issues arose primarily due to NeurIPS page limits, and we appreciate the feedback. All these points will be addressed in the revision.

$\quad$

## Complexity analysis
>We appreciate reviewer bZ6q’s constructive suggestion about having a complexity analysis. We summarize the analysis here and will add it in the revised version.
Given a multi-relational database $G = (R, E)$ with $m$ tables, $n$ foreign key constraints, and $p$ rows per table. Given a p-row table, we denote the time complexity of performing GMM clustering $c_{GMM}(p)$, training a diffusion model as $c_{diff}(p)$, training a classifier as $c_{class}(p)$, synthesizing as $c_{syn}(p)$, ANN searching complexity as $c_{ann}(p)$.

>**Phase 1: Latent Learning and Table Augmentation:**
>- Runtime: $n \cdot c_{GMM}(p)$.

>**Phase 2: Training:**
>- Runtime: $n \cdot c_{class}(p) + m \cdot c_{diff}(p)$.
>- This phase is dominated by diffusion training, primarily influenced by $m$.

>**Phase 3: Synthesis:**
>- Runtime: $n \cdot c_{syn}(p)$.

>**Additional Step: Matching:**
>- Runtime: $n \cdot c_{ann}(p)$.
>- Negligible runtime with FAISS implementation.

>**Summary:**
>- Dominated by Phase 2 (training) and Phase 3 (synthesis).
>- Critical factors: $m, n$, and $p$.
>- Robust against the number of clusters $k$ in Phase 1 due to the dominance of later phases.
Our model shows significant scalability and practicality compared to existing methods like SDV, which is limited to synthesizing at most 5 tables with a depth of 2 (section 5.2). We will provide empirical runtime measurements and baseline comparisons in the revised version.

$\quad$

## Experiments

>Multi-table synthesis, as a superset of single-table synthesis, necessitates more comprehensive evaluations, especially to assess how well the synthetic dataset captures inter-table correlations. Current metrics (e.g., machine learning efficacy, statistical metrics used in TabSyn and TabDDPM) are insufficient. In addition to including experiment results for the previous metrics, we have proposed a novel multi-table quality measure: long-term dependency, which evaluates the capture of long-range correlations in a multi-table database. Reviewers noted the lack of clear demonstration and formulation of this metric. We agree and will include an additional section detailing this metric.

>Furthermore, reviewers suggested adding a finer-grained ablation study on the number of clusters k. We conducted these experiments and discovered meaningful trends that can guide practical hyperparameter searches. We appreciate this suggestion and will incorporate the findings in the revised version.

$\quad$

## Limitation and future work

>As pointed out by reviewers, we are missing a discussion about limitations and future works. This is because of the page limit. We are going to add the following discussion in the revised version:

>“We focused on foreign key constraints in this work, and made the assumption that child rows are conditionally independent given corresponding parent rows. This brings three natural follow-up research directions: i) extension to the scenarios where this prior information is not available and these relationships need to be discovered first[3], ii) further relaxing the assumptions, and iii) inspecting multi-relational data synthesis with other integrity constraints (e.g, denial constraints[4], general assertions for business rules).
Furthermore, we evaluated ClavaDDPM's privacy with the common (in tabular data literature) DCR metric. Nonetheless, we think it is worthwhile to: i) evaluate the resiliency of ClavaDDPM against stronger privacy attacks[5], and ii) investigate the efficacy of boosting ClavaDDPM with privacy guarantees such as differential privacy. Similarly,  the impacts of our design on fairness and bias removal, as another motivating pillar in synthetic data generation, is well worth exploring as future work. We believe the thorough multi-relational modeling formulation we presented in this work, can serve as a strong foundation to build private and fair solutions upon. ”

---

### Author Response · Authors · 2024-08-06
**Reference List and Table 2**

**Reference**

[1] K. Cai, X. Xiao, and G. Cormode. Privlava: synthesizing relational data with foreign keys under differential privacy. Proceedings of the ACM on Management of Data, 1(2):1–25, 2023.

[2]N. Patki, R. Wedge, and K. Veeramachaneni. The synthetic data vault. In 2016 IEEE international conference on data science and advanced analytics (DSAA), pages 399–410. IEEE, 2016.

[3] F. Li, B. Wu, K. Yi, and Z. Zhao. Wander join: Online aggregation via random walks. In Proceedings of the 2016 International Conference on Management of Data, pages 615–629,2016.

[4] C. Ge, S. Mohapatra, X. He, and I. F. Ilyas. Kamino: constraint-aware differentially private data synthesis. Proc. VLDB Endow., 14(10):1886–1899, jun 2021.

[5] T. Stadler, B. Oprisanu, and C. Troncoso. Synthetic data – anonymisation groundhog day. In 31st USENIX Security Symposium (USENIX Security 22), pages 1451–1468, Boston, MA, Aug. 2022. USENIX Association.

[6] A. Kotelnikov, D. Baranchuk, I. Rubachev, and A. Babenko. Tabddpm: Modellingtabular data with diffusion models. In International Conference on MachineLearning, pages 17564–17579. PMLR, 2023.

[7] H. Zhang, J. Zhang, B. Srinivasan, Z. Shen, X. Qin, C. Faloutsos, H. Rangwala, and G. Karypis. Mixed-type tabular data synthesis with score-based diffusion in latent space. arXiv preprint arXiv:2310.09656, 2023.

$\quad$

**Table 2: ClavaDDPM vs ClavaTabsyn**

|Berka|Overall|Col Shape|0-hop|Cardinality|1-hop|
|-|-|-|-|-|-|
|ClavaTabsyn|93.39|88.65|92.83|93.47|98.59|
|ClavaDDPM|96.29|94.56|94.12|97.55|98.93|

---

### Comment · Area_Chair_5E22 · 2024-08-07
**Please read and respond to the rebuttal**

Dear reviewers,

First of all, thank you for your service to the ML community. Writing high-quality reviews and engaging with authors' responses is essential to a healthy ML research community.

It is essential that you read the rebuttals and provide a response and/or follow-up questions within the next few days. This will allow the authors sufficient time to react. While a detailed response addressing all points is not necessary, at a minimum, you should indicate you have read and considered the review and whether you will maintain or revise your score. Please also take the time to read the other reviews.

I want to thank you again, and I look forward to following the discussions here.

---

### Decision · Program_Chairs · 2024-09-25

**Decision:**

Accept (poster)

**Comment:**

Despite the relatively low mean score of the paper, I recommend the paper for acceptance, albeit with less certainty. The reason for this decision is that one of the reviewers was very positive about the paper and provided a set of strong arguments in favor of the proposed method:

- Important problem: Synthetic data generation is growing in the era of large, data-hungry models whose quality depends on the quality of training data. Moreover, an application of this problem to relational databases.
- Novelty: ClavaDDPM is the first tabular DDPM that supports generation conditioned on multiple tables (organized in several hierarchy levels) and can be used on real-world datasets.
- Innovative: The idea of learning latent variables with classifier guidance for grouped keys is interesting. In addition, the model supports both numerical and categorical data and includes parent-child relationships.
- Strong empirical results: ClavaDDPM was compared against existing non-diffusion multi-table models (PrivLava and Synthetic Dava Vault), the authors also adapted single-table baselines CTGAN and TabDDPM in several versions as baselines. High-quality ablation studies.
- Well-written paper.

One reviewer (bZ6q) provided a rather shallow review, likely written with the help of an LLM, and responded to the rebuttal with comments that I could not comprehend. The other negative reviewer (5kZH) ghosted the authors after the rebuttal was submitted despite several attempts to obtain a response. In my opinion, the authors provided reasonable responses in their rebuttal to said reviewer. For instance, the reviewer simply mentioned other methods to compare to without explaining why comparing to these methods adds any value to the paper. Hence, I think that these two reviews do not meet the quality standards of NeurIPS. Hence, not taking these into account, one reviewer is very positive about the paper, and one gives it a score of 5. Overall, this points to accepting the paper. I also think that his paper falls into the "new work" category (a new perspective on an important problem), which often receives harsher criticism than papers that fall into more established "mainstream" areas. For all of these reasons, I recommend to accept the paper.